# PEBP balances apoptosis and autophagy in whitefly upon arbovirus infection

Shifan Wang 📷 1,2, Huijuan Guo1,2, Keyan Zhu-Salzman3, Feng Ge 📷 1,2,4✉ & Yucheng Sun 📷 1,2✉

Apoptosis and autophagy are two common forms of programmed cell death (PCD) used by host organisms to fight against virus infection. PCD in arthropod vectors can be manipulated by arboviruses, leading to arbovirus-vector coexistence, although the underlying mechanism is largely unknown. In this study, we find that coat protein (CP) of an insect-borne plant virus TYLCV directly interacts with a phosphatidylethanolamine-binding protein (PEBP) in its vector whitefly to downregulate MAPK signaling cascade. As a result, apoptosis is activated in the whitefly increasing viral load. Simultaneously, the PEBP4-CP interaction releases ATG8, a hallmark of autophagy initiation, which reduces arbovirus levels. Furthermore, apoptosis-promoted virus amplification is prevented by agonist-induced autophagy, whereas the autophagy-suppressed virus load is unaffected by manipulating apoptosis, suggesting that the viral load is predominantly determined by autophagy rather than by apoptosis. Our results demonstrate that a mild intracellular immune response including balanced apoptosis and autophagy might facilitate arbovirus preservation within its whitefly insect vector.

[1] State Key Laboratory of Integrated Management of Pest Insects and Rodents, Institute of Zoology, Chinese Academy of Sciences, Beijing 100101, China. [2] CAS Center for Excellence in Biotic Interactions, University of Chinese Academy of Sciences, Beijing 100049, China. [3] Department of Entomology, Texas A&M University, College Station, TX 77843, USA. [4] Institute of Plant Protection, Shandong Academy of Agriculture Sciences, Jinan 250100, China. ✉email: gef@ioz.ac.cn; sunyc@ioz.ac.cn

Programmed cell death (PCD), including apoptosis and autophagy, is intrinsically connected with immunity and host defense against pathogen infection[1–7]. Autophagy is an intracellular degradation system involving sequestration of portions of cytoplasm by autophagosomes formed by fusion of double-membrane vesicles with lysosomes, in which autophagic cargo is degraded[8,9]. This lysosomal degradation mechanism contributes to cell differentiation, development, and homeostasis[10]. Apoptosis, the most extensively studied PCD, can be triggered through intrinsic and extrinsic pathway where a number of caspases that belong to cysteine proteases are activated[10,11]. Caspases then activate catabolic hydrolases, capable of degrading most cellular macromolecules in a controlled manner[12]. Distinct features of apoptotic cells include cytoplasmic shrinking, cell rounding, chromatin condensation, DNA fragmentation and membrane blebbing[3,5,7]. While PCD-associated degradation effectively protects hosts from pathogen invasion, it usually causes certain tissue damage[13]. By contrast, insect vectors seem to be more compatible with arthropod-borne virus (arbovirus) as low or no pathological symptoms are seen during virus acquisition and transmission[14–17]. Evidence suggests that arbovirus-induced PCD in insect vectors permits coexistence of arbovirus and the vector[18–20]. The molecular and biochemical bases, however, remain largely elusive.

Controlled cell destruction is often considered as an innate defensive mechanism to counteract viral infection. Many viruses suppress apoptosis to prevent premature host cell death and thus enhance virus proliferation[5]. On the other hand, activation of apoptosis in insect vectors is usually thought to be beneficial to arbovirus. For instance, suppression of apoptosis in *Aedes aegypti* decreases virus load and impairs virus dissemination[21]. Conversely, despite damage to gut barrier and modification of cellular regeneration, activation of apoptosis in mosquito midgut allows viruses to reach the vector's hemolymph[22,23]. Crosstalk between apoptosis and autophagy also impacts host infection[24,25]. Although autophagic cell death (or type II cell death) is an alternative pathway leading to demise of target cells, autophagy itself normally suppresses apoptosis to constitute a stress adaption that avoids cell death[24,25]. In contrast to apoptosis, arbovirus-induced autophagy helps host cells to resist arbovirus infection and replication. This intracellular resistance mechanism is exemplified by tomato yellow leaf curl virus (TYLCV) that is primarily transmitted by whitefly[26]. TYLCV activates autophagy in midgut and salivary glands of the Middle East Asia Minor 1 (MEAM1) whitefly species, and suppression of autophagy increases the TYLCV abundance[26]. In these examples, apoptosis and autophagy stimulated by arbovirus were studied separately. Considering the possible interplay, it is conceivable that apoptosis and autophagy could occur simultaneously and interact with each other spatially and temporally to maintain a balance between vector and virus load.

Host infection by arbovirus often is acute, but persistent arbovirus can coexist with vector for generations, especially those that can be transmitted via developing eggs. Vector insects are able to endure virus infection meanwhile minimize fitness cost by maintaining an optimal viral load[27–29]. Hence, a mild immune response could be the basis for virus tolerance[29,30]. For instance, mosquitoes with impaired apoptosis and autophagy reduce dengue virus infection[21]. In addition, overly activated immune system also causes production and accumulation of large amount cytokines, oxidative stress, and NO, resulting in autoimmunity and neurodegeneration[31]. Apoptosis and autophagy are the best studied intracellular immune responses in virus-infected vectors[14], but it is unclear how they shape vector's immune tolerance to arbovirus, and how arbovirus in turn manipulates immunity in the insect vector. Understanding such interaction, however, is crucial in control of vector-borne diseases [29,32].

The whitefly (*Bemisia tabaci*) is a notorious insect vector worldwide and transmits multiple DNA viruses belonging to Geminiviridae, the most devastating pathogens infecting hundreds of crops[33]. The MEAM1 and Mediterranean (MED) whitefly species transmit TYLCV, a monopartite begomovirus (family Geminiviridae) in a persistent-propagative manner[33]. Previous results showed that both apoptosis and autophagy could be activated by TYLCV in MEAM1 whitefly[20,26], and that virus load is enhanced by activation of apoptosis, but suppressed by initiation of autophagy[20,26]. Here, we showed that once bound by TYLCV CP, phosphatidylethanolamine-binding protein 4 (PEBP4) in whitefly interacted with Raf1, leading to suppressed phosphorylation of MAPK signaling components and thereby activated apoptosis. Meanwhile, binding to CP released ATG8 from PEBP4, which triggered the autophagy pathway. Balanced apoptotic and autophagic activities optimized arbovirus abundance and whitefly fitness. Our results revealed a function of PEBP4 in regulating whitefly immune homeostasis.

## Results

**TYLCV simultaneously induced apoptosis and autophagy in whitefly.** Since the midgut and the salivary gland are important barriers for arbovirus circulation within the insect vector, they were dissected to determine the induction dynamics of the apoptosis and autophagy processes in viruliferous whitefly. The TUNEL assay and immunofluorescence results showed that both apoptosis and autophagy were activated in the midgut and salivary gland at 24 h post-infection (hpi), although basal apoptosis and autophagy was also observed in salivary gland (Fig. 1a, b). Viral acquisition decreased expression of the anti-apoptotic genes *Iap* and *Bcl-2*, but increased expression of the pro-apoptotic genes *Caspase1* and *Caspase3* (Fig. 1c). Furthermore, the autophagy-related genes *ATG3*, *ATG8*, *ATG9*, and *ATG12* were also up-regulated in viruliferous whitefly (Fig. 1d). Consistently, activation of apoptosis and autophagy at the protein level were further confirmed by the time-course immunoblotting assay. The full-length Caspase3 was rapidly converted to cleaved Caspase3 during 6 to 24 hpi, suggesting a rapid induction of apoptosis in viruliferous whitefly. Likewise, remarkable consumption of the autophagy substrate SQSTM1 and drastic increase in ATG8 lipidation (conjugated to phosphatidylethanolamine, ATG8-PE) strongly suggested a simultaneous induction of autophagy and apoptosis (Fig. 1e).

**PEBP4 directly interacted with the coat protein (CP) of TYLCV.** Since CP has been shown to be required for successful infection[34], it was used as the bait in yeast two-hybrid (Y2H) to screen for target proteins in whitefly that may potentially activate apoptosis and autophagy. Twenty-three nonredundant sequences were firstly blasted against the genome of MED whitefly (http://gigadb.org/dataset/100286) to acquire the full amino acid sequences, which were then annotated referring to the database of Hemiptera in NCBI (Supplementary Table 1). Of all the potential proteins, we selected a putative phosphatidylethanolamine-binding protein (PEBP) due to its possible function in PCD regulation[35–37]. The phylogenetic analysis and sequence alignment showed that this PEBP was homologous to the PEBP4 family members, with a conserved PE-binding domain, and hereafter was named as whitefly PEBP4 (Supplementary Fig. 1).

Interaction between TYLCV CP and PEBP4 was re-verified using one-to-one Y2H analysis. Growth was observed in colonies that harbored both proteins when plated on the triple dropout (TDO) medium, but not on more stringent quadruple dropout (QDO) medium (Fig. 2a). Two custom polyclonal PEBP4 antibodies against different synthetic peptides were used in the

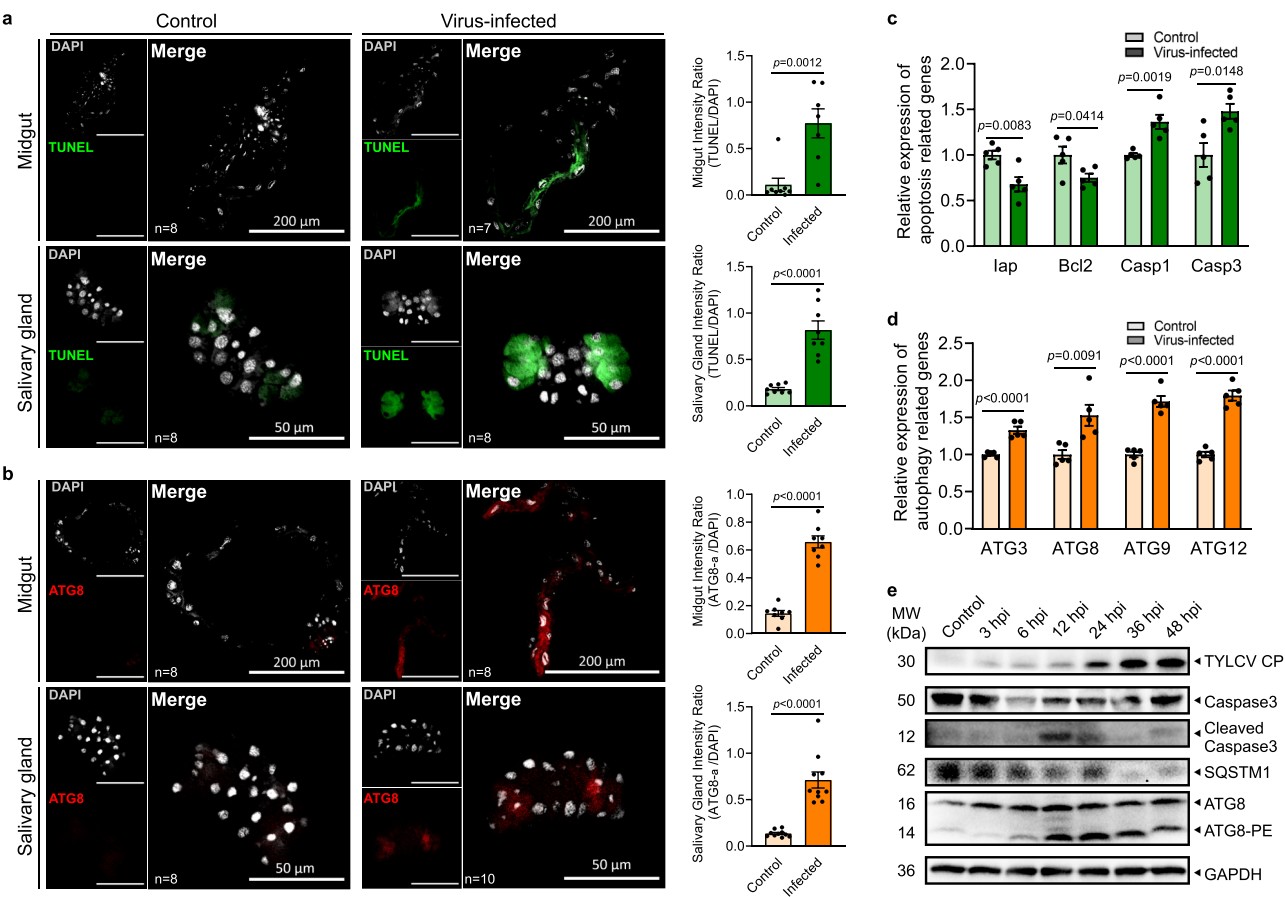

**Fig. 1 Apoptosis and autophagy were simultaneously induced in whitefly when acquiring TYLCV.** Midguts and salivary glands were dissected from nonviruliferous and viruliferous whitefly, respectively. **a** Apoptosis was determined by TUNEL assays (green), and **b** autophagy was indicated by the hallmark ATG8-PE (red). The nuclei (white) were stained by DAPI. Sample sizes (*n*) for statistical tests indicated in the panels refer to biologically independent whitefly. Relative intensity was quantified by ImageJ. Whiteflies feeding on TYLCV-infected plant for 24 hours were sampled. The relative expressions of **c** anti-apoptotic genes (*Iap*, *Bcl2*), pro-apoptotic genes (*Casp1*, *Casp3*), and **d** autophagy-related genes (*ATG3*, *ATG8*, *ATG9*, *ATG12*) were determined by qPCR. Five independent samples were used for each treatment. Values in bar plots represent mean ± SEM. All data were checked for normality by the Wilk-Shapiro test. Two-sided paired *t*-test was used to separate the means of normally distributed data, while Mann-Whitney test was used to analyze nonparametric distributed data, no multiple comparisons were performed in each test. **e** Time-course immunoblot monitored the dynamics of apoptosis and autophagy activation in whole body of whitefly, and GAPDH served as the loading control. The consumption of SQSTM1, a crucial autophagy substrate, represented autophagy activation. ATG8-PE represented lipidated ATG8, the hallmark of autophagy. PE phosphatidylethanolamine.

co-immunoprecipitation (Co-IP) assays to corroborate the CP-PEBP4 interaction in vivo. TYLCV CP was co-immunoprecipitated with PEPB4 in viruliferous whitefly lysate and not non-viruliferous whitefly lysate (Fig. 2b). To determine domain-specificity, whitefly PEBP4 was expressed as two peptide fragments: one contained a species-specific pre-domain and the other covered a conserved PE-binding domain. Pull-down assays indicated that PEBP4 interacted with the virus CP through its pre-domain and such interaction possibly was MED whitefly-specific (Fig. 2c).

Immunofluorescence showed that PEBP4 was abundant near the cell surface (Supplementary Fig. 2). In viruliferous whitefly, CP was found colocalized with PEBP4 in midgut and salivary gland cell membrane (Fig. 2d). Z-Stack scanning also showed that this interaction occurred in cytoplasm, in addition to plasma membrane (Fig. 2d). These results further confirmed direct CP-PEBP4 interaction.

**PEBP4 knockdown in viruliferous whitefly attenuated TYLCV-induced apoptosis but enhanced autophagy.** Abundance of PEBP4 at the protein level decreased initially upon TYLCV acquisition (0–6 hpi), and remained consistent from 6 to 48 hpi

(Supplementary Fig. 3a), whereas its mRNA increased after 48 hpi (Supplementary Fig. 3b). One scenario could be that PEBP4 protein was constantly captured because new virions were continuously ingested by whitefly. Feeding on *dsPEBP4* decreased *BtPEBP4* expression in whitefly (Fig. 3a). Knockdown of *BtPEBP4* increased gene expression of anti-apoptotic *Iap* and *Bcl2* but reduced pro-apoptotic *Caspase1* and *Caspase3*, indicating that apoptosis in whitefly was positively regulated by PEBP4 (Fig. 3b). By contrast, autophagy-related genes *ATG3* and *ATG12* were induced when whitefly fed on *dsPEBP4*, indicating PEBP4 negatively regulated autophagy (Fig. 3c). The opposing impact of PEBP4 on the two PCD processes was further supported by the fluorescence data (Fig. 3d, e) as well as the immunoblot showing increased Caspase3 cleavage, SQSMT1 consumption and ATG8 lipidation, and decreased TYLCV load as a result of *BtPEBP4* knockdown (Fig. 3f). Taken together, apoptosis and autophagy in whitefly could be independently regulated by PEBP4-CP interaction.

**PEBP4 interacted with Raf1 and negatively regulated MAPK pathway phosphorylation to enhance the virus-induced apoptosis.** The MAPK phosphorylation pathway Raf1/MEK/ERK

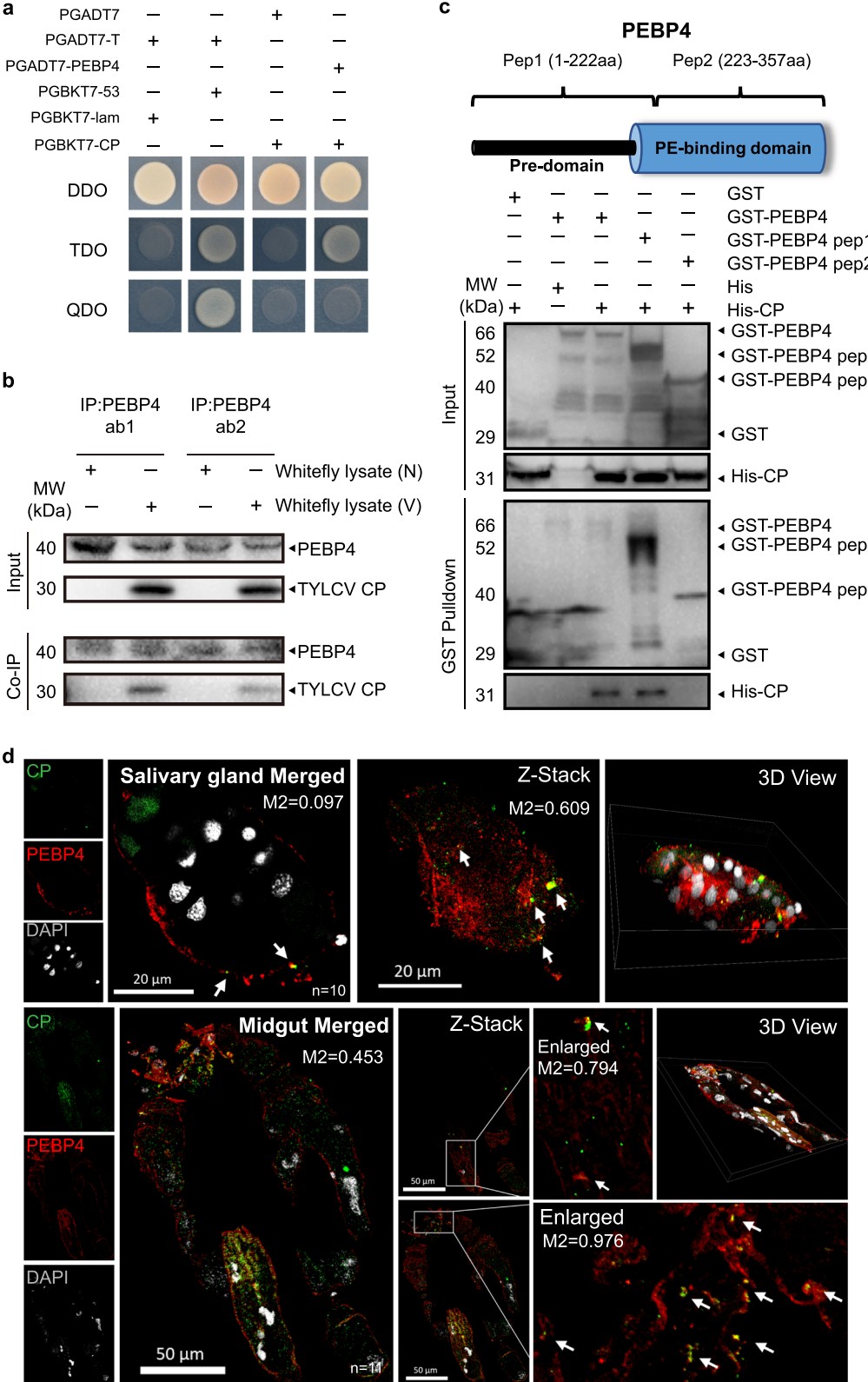

relays PEBP regulatory information on apoptosis[36,37]. All members of the human PEBP family are capable of regulating the MAPK cascade[38–42]. To investigate the role of whitefly PEBP4 in TYLCV-induced apoptosis, we examined if there existed direct interaction between PEBP4 and Raf1, in vivo and in vitro. GST pull-down and Co-IP assays showed that PEBP4 interacted with Raf1 via its conserved PE-binding domain whether CP was present or not (Fig. 4a, b). As virus acquisition continued, whitefly increased unphosphorylated Raf1 and decreased phosphorylated Raf1, ERK and MEK were observed (Fig. 4c). Since increased transcription (Supplementary Fig. 4a) could complicate interpretation of elevated unphosphorylated Raf1, we examined phosphorylation by whitefly total protein lysate at increasing GST-PEBP4 and His-CP concentrations respectively. Raf1

**Fig. 2 PEBP4 directly interacted with the CP of TYLCV. a** PEBP4 interacted with TYLCV CP in yeast two-hybrid assay. Lane 1: negative control, lane 2: positive control, lane 3: blank, lane 4: PEBP4-CP interaction. **b** Co-IP assay. Total proteins of the viruliferous (V) vs. non-viruliferous (N) whitefly were extracted. Two polyclone antibodies (ab1, ab2) of whitefly PEPB4 were used. **c** GST pull-down assays of full-length PEBP4, pep1 (consisting of a whitefly specific pre-domain) and pep2 (containing a conserved PE-binding domain). Recombinant His-CP interacted with both GST-PEBP4 and GST-PEPB4 pep1, but not GST-PEPB4 pep2. GST and His tags expressed by empty vectors were used as negative control. **d** PEPB4 (red) was co-localized with CP (green). Nuclei (white) were stained by DAPI. Sample sizes (*n*) for statistical tests indicated in the panels refer to biologically independent whitefly. Arrows mark merged signals. Colocalization was quantified by ImageJ Coloc2 using Manders' Colocalization Coefficients (MCC). M2 values represented the proportion of PEBP4-CP colocalization of total CP. All tissues were collected from 24 hpi whitefly.

phosphorylation was attenuated by both GST-PEBP4 and TYLCV CP (Fig. 4d, e). This biochemical evidence and knockdown experiment (Supplementary Fig. 4b) clearly indicated that PEBP4 is a negative regulator of MAPK signaling pathway. Therefore, elevated transcription and dephosphorylation could explain abundant unphosphorylated Raf1 resulted from TYLCV infection. Interestingly, Co-IP with gradient CP or Raf1 did not reveal any competition for PEBP4 binding between them in viruliferous whitefly (Fig. 4f). Consistently, PEBP4-Raf1 interaction in both midgut and salivary gland (Supplementary Fig. 4c) apparently could involve viral CP, in the form of the CP-PEBP4-Raf1 triple complex (Fig. 4g).

MAPK pathway has been shown to negatively regulate apoptosis[24,25,43]. ERK-regulated anti-apoptotic genes, including *Iap* and *Bcl-2*, counteract with pro-apoptotic genes[10,11]. Suppression of anti-apoptotic genes could increase the ratio of pro-apoptotic genes products and eventually induced apoptosis[24,25,43,44]. We hypothesized that phosphorylation of MAPK pathway negatively regulated the TYLCV-induced apoptosis, and vice versa. The immunofluorescence and immunoblot results showed that TYLCV-induced apoptosis was enhanced by Mirdametinib, an inhibitor of MEK phosphorylation (Fig. 4h and i). Although it does not directly inhibit Raf1, Mirdametinib decreased unphosphorylated Raf1, possibly due to the feedback regulation of MAPK pathways[45]. Presumably, the presence of TYLCV CP stabilized PEBP4 binding to Raf1, which suppressed the phosphorylation of MAPK signaling cascade and promoted apoptosis (Fig. 4j).

**TYLCV CP competitively interacted with PEBP4 to liberate ATG8 and promoted autophagy.** Human PEBP1 interacts with PE-unconjugated LC3B, the microtubule-associated protein 1 light chain 3 β, at the LC3-interacting region (LIR) motif to suppress autophagy in a MAPK pathway-independent manner[35]. Once phosphorylated at Ser153, PEBP1 dissociates from the PEBP1-LC3B complex and induces autophagy[35]. We found that whitefly PEBP4 also contained putative LIR motifs according to the iLIR Autophagy Database (https://ilir.warwick.ac.uk/index.php) (Supplementary Fig. 5). GST-pulldown and Co-IP assays demonstrated that whitefly PEBP4 interacted with ATG8, the LC3 homolog in insects, both in vitro and in vivo (Fig. 5a, b). The in vitro co-incubation assays showed enhanced autophagy as recombinant CP increased (Fig. 5c). By contrast, whitefly lysate co-incubated with PEBP4 had little effect on autophagy without TYLCV induction (Fig. 5d), suggesting that in nonviruliferous whitefly ATG8 sequestered by PEBP4 was unable to activate autophagy. In the presence of CP, however, CP-PEBP4 interaction freed ATG8, activating autophagy. Excess PEBP4 impeded ATG8 liberation, and thereby prevented virus-induced autophagy (Fig. 5e). Presumably, virus CP could competitively interact with PEBP4 to dissociate PEBP4-ATG8 interaction. A competitive pull-down assay showed that excess ATG8 had little effect on PEBP4-CP interaction while excess of virus CP impaired the PEBP4-ATG8 association (Fig. 5f). It appeared that PEBP4 had higher affinity for TYLCV CP than for ATG8. Results were

further confirmed by immunofluorescence using the ATG8 antibody that also labels unconjugated ATG8. In nonviruliferous whitefly, ATG8 colocalized with PEBP4 on membrane (Supplementary Fig. 5c). However, after acquiring TYLCV, viral CP only bound with PEBP4 and not ATG8 (Fig. 5g), indicated that these three proteins were unable to form a complex. Taken together, these findings suggest that autophagy was inactivated in nonviruliferous whitefly because ATG8 was arrested by PEBP4. Once infected by TYLCV, the virus CP competitively interacted with PEBP4 and liberated ATG8 that supposedly underwent lipidation and initiated the formation of autophagosome (Fig. 5h).

**A mild PCD response ensured coexistence of TYLCV and whitefly.** It has been shown that TYLCV load is promoted by apoptosis but suppressed by autophagy in MEAM1 whitefly[20,26], but the role of simultaneous activation of apoptosis and autophagy in whitefly co-existing with TYLCV is mysterious. We treated whiteflies with a series of pharmacological compounds that influence these PCD processes, individually and in combination, to determine their effects on virus load. Results showed that autophagy inhibitor 3-MA and apoptosis agonist PCA-1 promoted viral load whereas autophagy agonist rapamycin and apoptosis inhibitor Z-VAD-FMK reduced it (Fig. 6a and b), supporting previous studies. Furthermore, agonist-induced apoptosis negatively regulated autophagy, evidenced by accumulation of autophagy substrate SQSTEM1, and increased virus load in whitefly. Conversely, inhibition of apoptosis reduced virus load (Fig. 6c, d). On the other hand, agonist-induced or inhibitor-suppressed autophagy had no effect on virus load, rather, virus abundance was significantly influenced by the autophagy status (Fig. 6c, d). These results suggested that suppressed autophagy was required for promoting TYLCV load in whitefly.

It is likely that stronger autophagy reduced virus abundance but compromised the fitness of whitefly; meanwhile, stronger apoptosis, although favored virus load, could potentially be detrimental to whitefly. Given this assumption, we experimentally tested whether PCD functioned to maintain the intricate balance of whitefly fitness and virus load (Fig. 6e–g). Reduced whitefly survival rate was observed by either suppression of autophagy by 3-MA or over-activation of autophagy by co-application of rapamycin and Z-VAD-FMK (Fig. 6e). To rule out effect due to TYLCV infection, survival rates on a time course was determined on nonviruliferous whitefly with over-activated autophagy. Decreased survival when treated with both rapamycin and Z-VAD-FMK suggested that over-activation of autophagy was accompanied by physiological costs in whitefly (Fig. 6f). On the other side, when antiviral autophagy was suppressed by feeding inhibitor 3-MA, virus load exceeded the capacity of whitefly, also resulting in impairment of whitefly survivorship (Fig. 6g). Similar results were also found in RNAi treatments (Supplementary Fig. 6). Together, these findings suggested that autophagy over-activation and suppression were both harmful to the fitness of viruliferous whitefly. Only PCD with well-balanced apoptosis and autophagy could ensure coexistence of TYLCV and whitefly without obviously physiological cost (Fig. 6h).

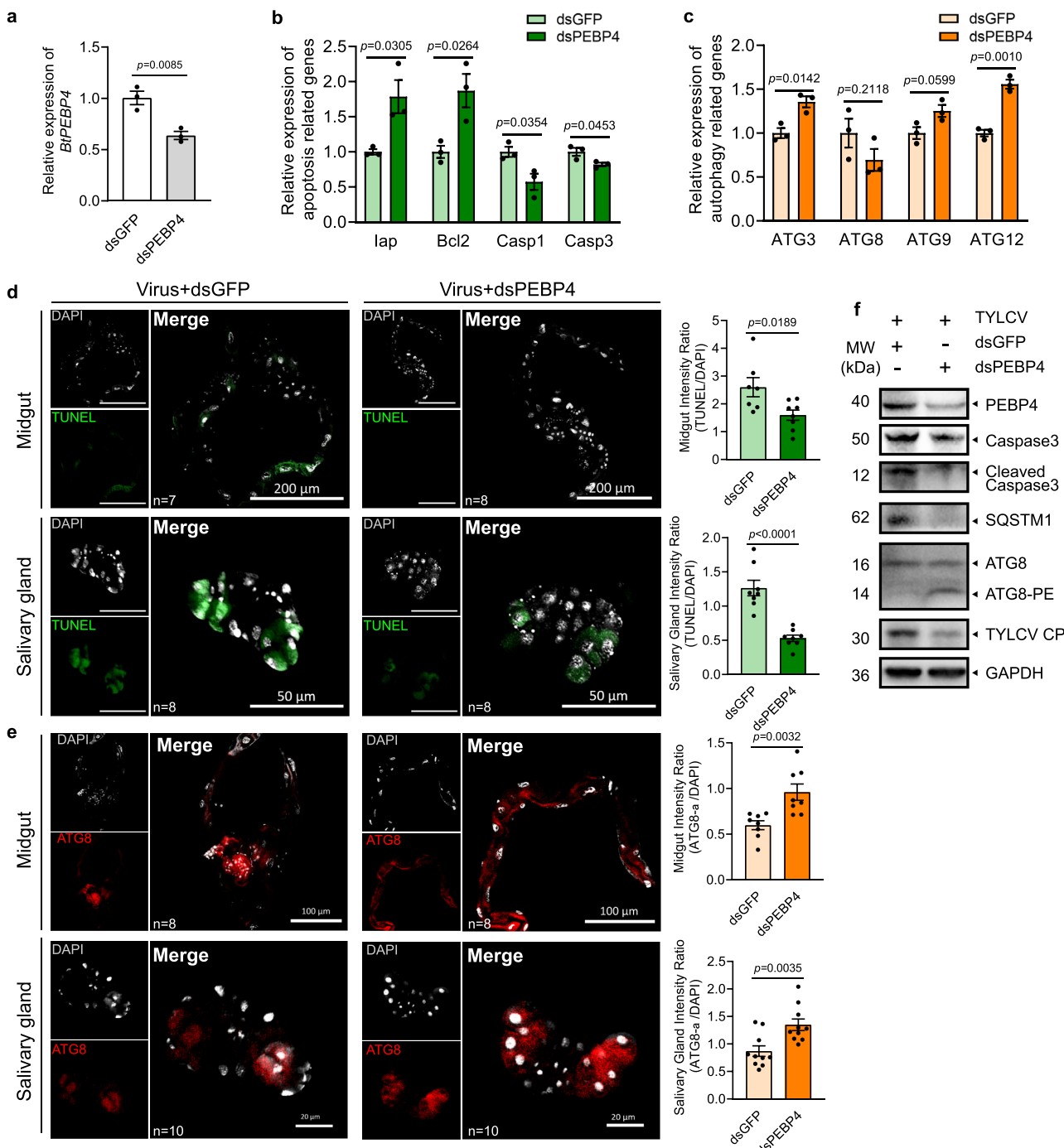

**Fig. 3 Knockdown of PEBP4 attenuated TYLCV-induced apoptosis but enhanced autophagy. a** *BtPEBP4* RNAi knockdown in whitefly reduced the expression of *BtPEBP4* at mRNA level, and *dsGFP* served as control. **b** The relative expression of apoptotic related genes (*Iap*, *Bcl2*, *Caspase1*, *Caspase3*), and **c** autophagy related genes (*ATG3*, *ATG8*, *ATG9*, *ATG12*) were determined using qPCR. Three independent samples were used for each treatment. All data were checked for normality by the Wilk-Shapiro test. Two-sided paired *t*-test was used to separate the means of normally distributed data, while Mann–Whitney test was used to analyze nonparametric distributed data, no multiple comparisons were performed in each test. **d** Apoptosis in midgut and salivary gland was determined by TUNEL labeling (green). **e** Autophagy was noted by ATG8-PE (red), and nuclei were stained by DAPI (white). Sample sizes (*n*) for statistical tests indicated in the panels refer to biologically independent whitefly. Relative intensity was quantified by ImageJ. Values in bar plots represent mean ± SEM. **f** Immunoblotting using whole body. GAPDH served as the loading control, and three independent biological replicates were conducted.

## Discussion

PEBPs have been well characterized for their regulatory function in host immunity that involves multiple signaling pathways, such as MAPK, TBK, NFκB, and GSK[41,46,47]. In the current study, we identified a PEBP in MED whitefly that served as a master regulator of apoptosis and autophagy in vivo. Coordination of the two PCD processes has likely optimized virus load and vector fitness, enabling virus-vector coexistence. Genomics analyses revealed 202 PEBPs in whitefly genome, which is in great contrast with 15 other arthropod species; each has 16 PEBPs or less[48].

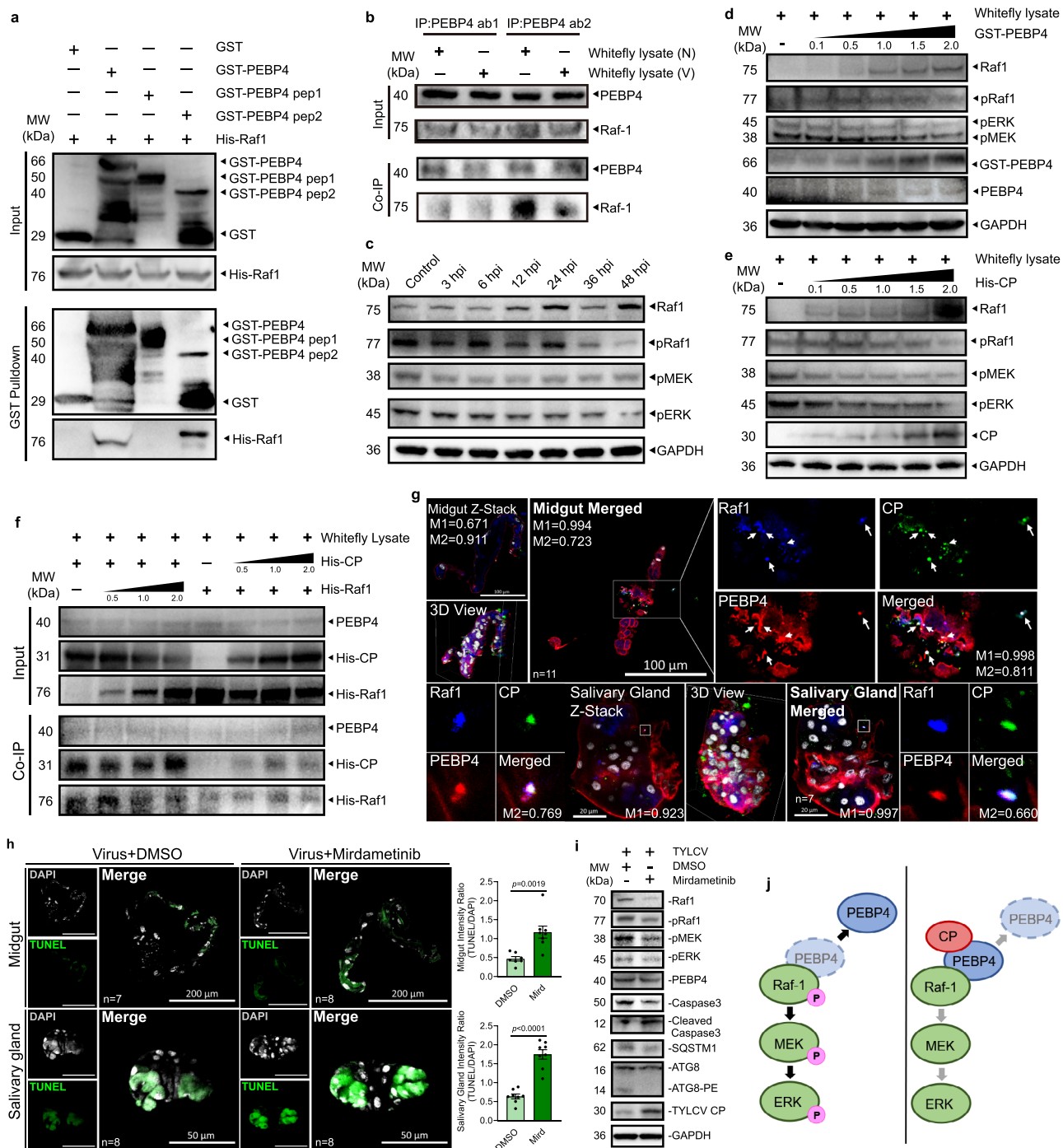

**Fig. 4 PEBP4 interacted with Raf1 which negatively regulated MAPK phosphorylation and enhanced the virus-induced apoptosis. a** GST pull-down showing the interaction between recombinant His-Raf1 and PE-binding domain of GST-PEBP4. **b** PEBP4 co-immunoprecipitated with Raf1 in viruliferous (V) or nonviruliferous (N) whitefly. **c** Time-course immunoblots of MAPK pathway phosphorylation. **d**, **e** Immunoblots of MAPK pathway phosphorylation in whitefly lysate when co-incubated with a concentration gradient of **d** PEBP4 and **e** TYLCV CP. **f** His-CP and His-Raf1 of different concentrations were co-incubated in whitefly lysate, and co-immunoprecipitated with PEBP4. **g** Raf1 (blue), PEBP4 (red), and CP (green) were simultaneously labeled in immunofluorescence assays. Colocalization was quantified by ImageJ Coloc2 using Manders' Colocalization Coefficients (MCC). M1 = PEBP4-Raf1/Total Raf1; M2 = CP-Raf1/Total Raf1. **h**, **i** After 24 h feeing on mirdametinib, a phosphorylation inhibitor of MEK, whiteflies were transferred to virus-infected host plants for another 24 h to acquire TYLCV. The midgut and salivary gland were dissected for **h** TUNEL labeling (green). Nuclei were stained by DAPI (white). Sample sizes (*n*) for statistical tests indicated in the panels refer to biologically independent whitefly. Values in bar plots represent mean ± SEM. All data were checked for normality by the Wilk-Shapiro test. Two-sided paired *t*-test was used to separate the means of normally distributed data. **i** Immunoblot of phosphorylation of MAPK pathway and initiation of apoptosis. **j** Model of CP-PEBP4 regulation on MAPK cascade in whitefly. CP stabilized Raf1-PEBP4 interaction, which blocked the MAPK phosphorylation cascade.

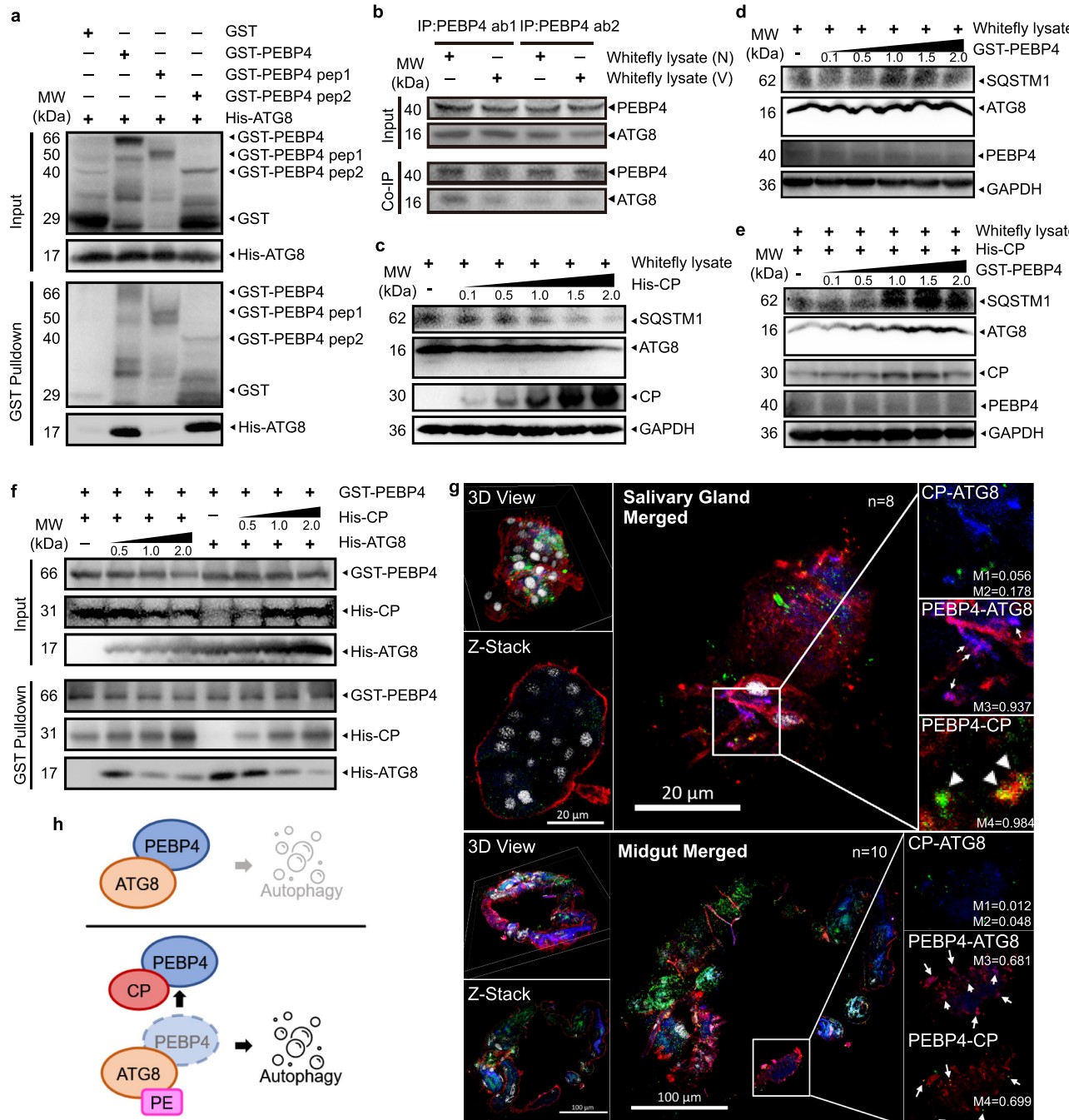

**Fig. 5 CP competitively interacted with PEBP4 to liberate ATG8 and promoted autophagy. a** GST pull-down showing the interaction between recombinant His-ATG8 and GST-PEBP4. **b** PEBP4 co-immunoprecipitated with ATG8 in viruliferous (V) or nonviruliferous (N) whitefly. **c–e** Recombinant His-CP and GST-PEBP4 with concentration gradients were co-incubated with the lysate of whitefly to monitor the activation of autophagy by immunoblotting, PEBP4 bands represent endogenous abundance. **f** Competition between CP and ATG8 for PEPB4 by competitive pull-down assays. His-CP and His-ATG8 were supplied within a concentration gradient to co-incubate with GST-PEBP4. **g** ATG8 (blue), PEBP4 (red), and CP (green) were simultaneously labeled and localized by immunofluorescence in viruliferous whitefly. The nuclei were stained by DAPI (white). Sample sizes (*n*) for statistical tests indicated in the panels refer to biologically independent whitefly. Colocalization was quantified by ImageJ Coloc2 using Manders' Colocalization Coefficients (MCC). M1 = CP-ATG8/Total ATG8; M2 = CP-ATG8/Total CP; M3 = PEBP4-ATG8/Total ATG8; M4 = CP-PEBP4/Toal CP. **h** Model of CP-PEBP4 regulated autophagy in whitefly. Unconjugated ATG8 arrested PEBP4 to prevent autophagy in nonviruliferous whitefly. Upon TYLCV invasion, CP competed for PEBP4 binding, which released ATG8 and initiated autophagy.

Transcriptome comparisons revealed higher expression of 20 PEBPs in whitefly populations with stronger virus transmission ability. Our study for the first time demonstrated the pivotal role of PEBP4 in modulating crosstalk between apoptosis and autophagy. Such PCD immune homeostasis is important in shaping the balance between virus loading and vector fitness. Our findings have deepened our understanding of complexed virus-insect vector interaction controlled by a single molecule from the insect vector, supporting the co-evolutionary theory of arbovirus and insect vectors in nature.

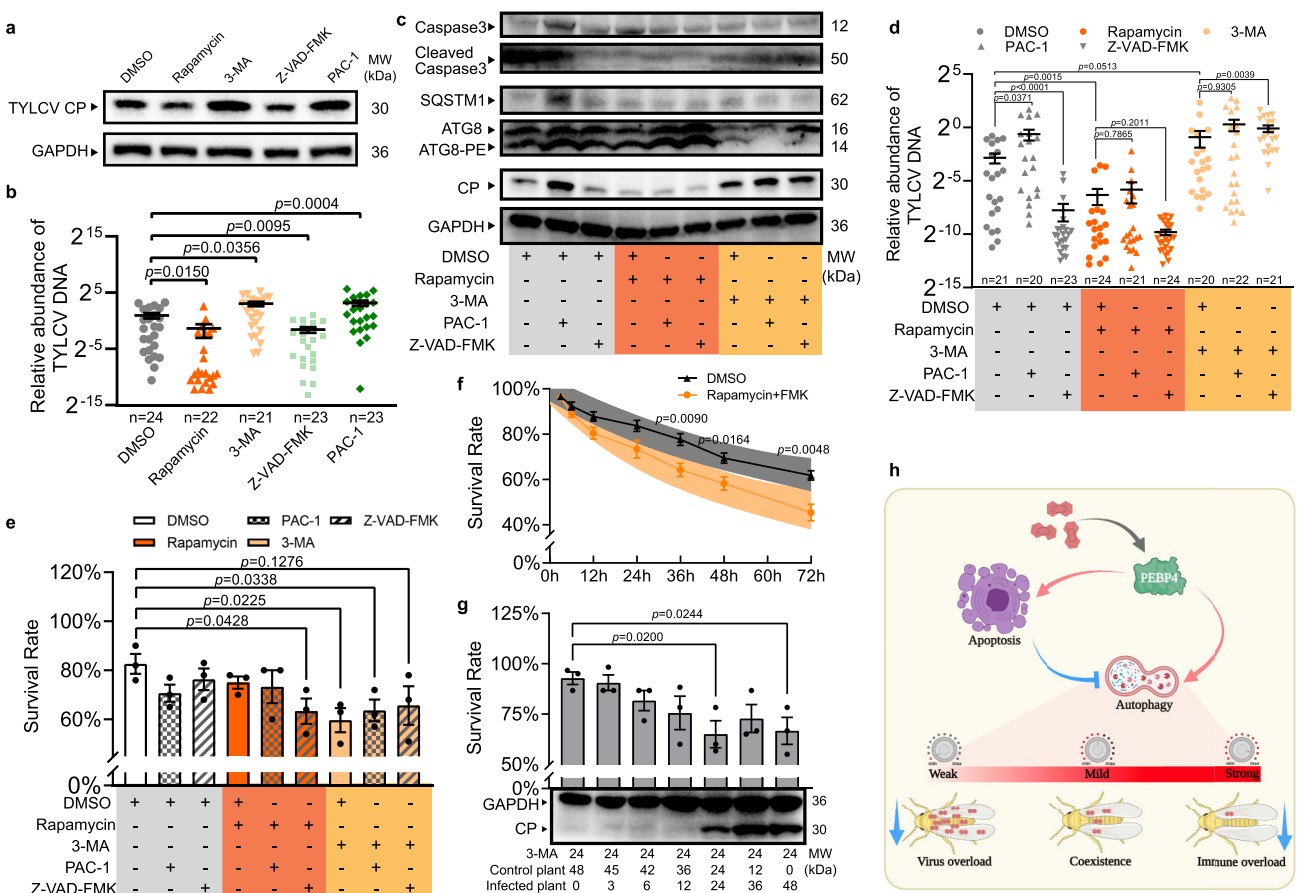

**Fig. 6 A mild intracellular immunity in whitefly facilitated its coexistence with TYLCV. a**, **b** Autophagy agonist rapamycin, autophagy inhibitor 3-MA, apoptosis agonist PAC-1, and apoptosis inhibitor Z-VAD-FMK were fed with 24 h in artificial diet respectively. **a** The relative protein abundance of TYLCV CP was determined by immunoblotting in viruliferous whitefly, and **b** the relative quantity of TYLCV DNA per whitefly was determined by qPCR. **c–e** Effects of agonist and inhibitor combinations on activation of apoptosis and autophagy in viruliferous whitefly, and on viral load and whitefly survival rate. Whiteflies were transferred to acquire virus for 24 h after agonist and/or inhibitor treatment via feeding. **c** The activation of apoptosis and autophagy in viruliferous whitefly was determined by immunoblotting, and **d** relative virus DNA was determined by qPCR. Sample sizes (*n*) for statistical tests indicated in the panels refer to biologically independent whitefly. **e** Whiteflies (100/replicate for each pharmacological combination treatment) were transferred to virus-infected plants for 48 h prior to survival rate measurement, three independent samples were used for each treatment. **f** Nonviruliferous whiteflies fed with rapamycin and Z-VAD-FMK were sampled for survival rate measurement, and their prediction bands (95% confidential interval) were generated based on regression result using linear quadratic survival model, five independent samples were used for each treatment. **g** Viral burden on whitefly survival rate with 24 h treatment of autophagy inhibitor 3-MA. Whiteflies, in groups of 60, was transferred to uninfected plants, and then to TYLCV-infected plants for time periods as indicated. Survival was recorded for survival rate calculation, and determination of TYLCV CP abundance by immunoblotting, three independent samples were used for each treatment. Values in bars or line plots represent mean ± SEM. All data were checked for normality by the Wilk-Shapiro test. Two-sided paired *t*-test was used to separate the means of normally distributed data, while Mann-Whitney test was used to analyze nonparametric distributed data, no multiple comparisons were performed in each test. **h** Model of interaction between apoptosis and autophagy in viruliferous whitefly and consequence on survival rate and virus load: TYLCV-induced apoptosis negatively regulated autophagy to maintain a homeostasis in favor of the coexistence of TYLCV within whitefly. This graph was created with BioRender.com.

## Regulation of PEBP-MAPK signaling cascade on apoptosis activation.

PEBP1, also called Raf1 kinase inhibitor protein (RKIP) in human was first discovered to possess Raf1-inhibiting activity[49]. Others PEBPs, such as human PEBP4, have also been shown to suppress MAPK signaling cascade by interacting with Raf1[50,51]. Our results revealed similar function of PEBP4 in whitefly. Therefore, insect PEBPs also negative regulate MAPK-apoptosis. Our results further demonstrated that TYLCV CP stabilized PEBP4-Raf1 interaction to suppress MEK/ERK phosphorylation. Given the importance of phosphorylation status of PEBP in its regulation of downstream signaling, we presume that TYLCV may be able to block the phosphorylation site of PEBP4 by forming a CP-PEBP4-Raf1 complex, and thus prevent its dissociation from Raf1, as illustrated in human, phosphorylation on T42 or S153 could release PEBP1 from Raf1[52,53].

ERK was typically characterized as transcriptional suppressor of pro-apoptotic factors and positive regulator of anti-apoptotic proteins[43,44]. Our results demonstrated that silencing PEBP4 resulted in increased anti-apoptotic *Iap* and *Bcl-1* expression, indicating PEBP4 positively regulated apoptosis in whitefly. TYLCV CP augmented PEBP4 inhibition on MAPK pathway, and thereby reduced *Iap* and *Bcl-1* expression. Such negative transcriptional regulation explained TYLCV-activated apoptosis. In most cases, apoptosis in virus-infected hosts is induced by physical binding of extracellular signals to the transmembrane death receptor[54]. Alternatively, apoptosis could also be initiated via the mitochondria-dependent intrinsic pathway that mostly relies on the homeostasis between pro-apoptotic factors and anti-apoptotic factors[11]. PEBP4-dependent activation of apoptosis in viruliferous whitefly apparently belonged to the latter, which

broadly connected with other intracellular signals involved in differentiation, development, and homeostasis[10,11].

Varied on the regulatory pathway involved, arbovirus species and vector tissue, apoptosis may have contrasting effects on arbovirus load within insect vectors. The antiviral JNK pathway-activated apoptosis restricts the arboviruses Dengue, Zika and Chikungunya in salivary gland of *Aedes aegypti*, while our study exhibited that MAPK-dependent activation of apoptosis promoted TYLCV load in MED whitefly[55]. Likewise, apoptosis occurred in the midgut of *Ae. aegypti* increases Sindbis virus infection[56]. Intriguingly, even in the brain of MED whitefly, a caspase-dependent apoptosis is induced by TYLCV and results in neurodegeneration, which behaviorally promotes TYLCV dissemination among host plants[20]. Effects of apoptosis activation on arbovirus load can be complicated. Illustration of mechanistic roles and unique patterns of apoptosis activation among insect vectors will help us to achieve comprehensive understanding of how viruses escape from vector immunity.

**Direct and indirect regulation on autophagy**. Strong evidence has indicated that PEBPs are involved in regulating autophagy-associated degradation[35,57]. Human PEBP1 is reported to interact with the LC3-interacting region (LIR) motif of PE-unconjugated LC3B, the homolog of ATG8 in mammals. The complex prevents initiation of autophagy that is independent of MAPK signaling pathway[35]. Moreover, human PEBP4 is co-localized with the lysosome, suggesting its potential role in regulation of degradation[57]. Similarly, whitefly PEBP4 was predicted to contain LIR motifs, capable of sequestering ATG8. TYLCV appeared to bind with PEBP4 via its CP, causing ATG8 release and initiation of autophagy. Increased CP protein was accompanied by elevated PEPB4-dependent autophagy. This dose-dependent manner most likely would avoid unnecessary self-consumption in insect vectors[6,58]. Continuous feeding on virus-infected plant presumably cause viral burden, resulting in over-activation of autophagy and causing pathological lesions, and thereby increased whitefly mortality[59]. Suppressed PEBP4-ERK-apoptosis signaling by CP counteracted the effect of excessive autophagy. Such regulation is CP concentration-dependent, providing a molecular explanation for arbovirus preservation within whitefly.

Different pre-domain sequences of PEBP4 between whiteflies from MEAM1 and MED reasonably explained the results of Y2H and Co-IP, as well as failure to screen any PEBPs in MEAM1 species that could interact with TYLCV CP[60,61]. Although TYLCV could also induce PCD in MEAM1 species, the most striking difference is the time required for PCD initiation: 6–12 h for MED whitefly and 24–48 h for MEAM1 whitefly[20,26]. The MED species with PEBP4-dependent PCD may have higher virus capacity than the MEAM1 species with PEBP4-independent PCD[62]. This discovery may help explain the rapid replacement of MEAM1 by MED species in Asia because arbovirus transmission could bring potential benefits for vector fitness[63].

**Immune homeostasis favors virus-vector coexistence**. When challenged with pathogens, hosts are not only directly impaired by microbial invaders but usually cause cell and tissue damages due to overwhelming immunity[64,32]. To prevent unnecessary costs of the immune system, disease tolerance strategy is widely adopted by plants or mammals, which prefers elimination of deleterious effects produced by immunopathology rather than conferring direct immunity against infectious pathogens[64,32]. Complex crosstalk between apoptosis and autophagy is often manifested in reciprocal antagonism. Presumably, it facilitated shaping a homeostasis during removing superfluous cells for

tissue development or in determining cell fate upon virus infection[24,25,65]. PEBPs were found to regulate apoptosis and autophagy independently[37], but their role in shaping apoptosis-autophagy homeostasis has been ignored. In our study, although virus load was primarily determined by autophagy, PEBP4 regulated coactivation ensured involvement of apoptosis as well. Its dynamic antagonistic effect on autophagy would prevent virus elimination when autophagy was continuously activated by virus ingestion, and thus warrant the vectoral role of whitefly. On the other side, coactivation of both apoptosis and autophagy prevented the substantial fitness cost of whitefly, because suppressing autophagy by application of 3-MA amplified the burden of virulence[32]. Thus, a mild immunity is required for balancing arbovirus load and vector fitness.

In conclusion, we have shown that TYLCV can simultaneously activate apoptosis and autophagy in MED whitefly via interacting with a vector protein PEBP4. PEBP4 serves as a molecular switch to regulate both MAPK pathway-relayed apoptosis and ATG8-initiated autophagy. Abundance of TYLCV within whitefly is mainly determined by anti-viral autophagy, since autophagy arrests apoptosis that facilitates virus load. The PCD homeostasis most likely has furnished an immune tolerance, allowing TYLCV to persist in whitefly populations. These results provide an evolutionary insight into PCD regulations in a vector that is more compatible with arbovirus. Molecular understanding of immune homeostatic mechanism is of paramount importance for refining immunoregulation-based approaches for future control of virus transmission.

## Methods

**Insect, plant, and virus**. The infectious clone of *Tomato yellow leaf curl virus* isolate SH2 (GenBank accession no: AM282874) was generously provided by Professor Xueping Zhou (State Key Laboratory for Biology of Plant Diseases and Insect Pests, Institute of Plant Protection, Chinese Academy of Agricultural Sciences). Tomato plants (*Solanum lycopersicum* cv Moneymaker), the natural host of TYLCV, were used for virus acquisition. Plants at the 3–4 true leaves stage were inoculated with the infectious clone, and confirmed by both visual observation and PCR analysis. All plants were grown at 26–28 °C, with 60% relative humidity and a photoperiod of 16 h: 8 h (light/dark) cycle. The Mediterranean (MED) *Bemisia tabaci* (mtCOI GenBank accession no: GQ371165) was reared on cotton plants placed in insect-proof cages. New emerged adult whiteflies with mixed sex were randomly collected for the experiments.

**PCR and quantitative PCR (qPCR)**. Total DNA of plants or insects was extracted using the RoomTemp™ Sample Lysis Kit (Vazyme, P073) according to manufacturer's protocol. A 412 bp TYLCV fragment was PCR amplified using Taq PCR MasterMix (Tiangen, KT201) and primers V61 and C473 as previously described[66]. Total RNA of whitefly samples was extracted by TRIzol™ Reagent (Ambion, 15596018), and reverse transcribed using the FastQuant RT Kit with gDNase (Tiangen, KR106). cDNAs of *PEBP4*, *PEBP4* pep1, *PEBP4* pep2, *V1* (TYLCV CP), *Raf1*, and *ATG8* were amplified by PrimerSTAR MAX DNA Polymerse (Takara, R045A). RT-qPCR reactions using the PowerUp™ SYBR Green Master Mix were carried out on the QuantStudio 12 K Flex Real-Time PCR System (ABI) (ABI, A25742). Data were analyzed by the $2^{-\triangle\triangle CT}$ relative quantification method, and each biological replicate consisted of three technical replicates.

**Yeast two-hybrid assays**. The cDNA library of *B. tabaci* was constructed in plasmid pGADT7 (the prey), using the Clontech cDNA Library Construction kit (Takara, 630490) and transferred into yeast competent cells (strain Y187). The titer of the primary cDNA library was determined using colony number on plates, and insert size was examined by colony PCR. The full-length TYLCV *V1* (CP) gene was cloned into PGBKT7 (the bait). Recombinant pGBKT7-CP was used to transform yeast strain Y2HGold and then selected on the S.D./-Leu medium. Finally, the Y187 with prey plasmid and the Y2HGold with bait plasmid were screened with stress. Positive clones were selected on the triple dropout medium (TDO: S.D./-His/-Leu/-Trp) and quadruple dropout medium (QDO: S.D./-Ade/-His/-Leu/-Trp) with 3-aminotriazole (3-AT), followed by sequencing analysis and search against the genome database of MED whitefly by Local Blastx 2.7.1[67]. Translated sequences were compared to the *Hemiptera* database in NCBI by Blastp. To verify the CP-PEBP4 interaction, full length PEBP4 was ligated into pGADT7 and co-transformed into yeast strain Y2HGold with pGBKT7-CP. Co-transformation of pGBKT7-lam and pGADT7-T vectors was used as the negative control. The presence of transgenes in yeast cells was confirmed by double dropout medium (DDO:

SD/-Leu/-Trp). Triple and quadruple dropout media were used to assess the CP-PEBP4 interaction.

**Recombinant protein expression.** To construct bacterial expression vectors, the amplified CD regions of *PEBP4*, *PEBP4 pep1* and *PEBP4 pep2* were cloned into pGEX-4T-1. The amplified CD regions of *V1* (TYLCV CP), *Raf1*, and *ATG8* were cloned into pET28a using *Xho* I and *EcoR* I sites. *Escherichia coli* BL21 (Takara, 9126) and BL21 (DE3) (Vazyme, C504) cells were used for fusion protein expression. All constructs were verified by restriction digestion and DNA sequencing (BGI, China). Sequence of whitefly *PEBP4* and all primers were listed in Supplementary Table 2.

All recombinant plasmids (GST-PEBP4, GST-PEBP4 pep1, GST-PEBP4 pep2, His-CP, His-Raf1, His-ATG8) and empty plasmids (GST tag, 6×His tag) were transformed into *Escherichia coli* BL21 (Takara, 9126) or BL21 (DE3) (Vazyme, C504) cells and induced by 0.1 M IPTG for 4–16 h at 16–37 °C.

**Western blotting.** Two rabbit anti-PEBP4 polyclone antibodies were prepared by BGI against synthetic peptides CPRKVRSRKNKENMES and TRHETTRSRP-KNISPC. Primary anti-Raf1 (D220484), anti-pRaf1 (D151522), anti-pMEK (D155070), anti-pERK (D151580), anti-SQSTM1 (8025 S), anti-ATG8/GABARAP (13733 S), and anti-GAPDH (60004-1-Ig) antibodies were respectively purchased from BBI, CST, and Proteintech. The production of rabbit anti-BtCaspase3b polyclone antibody has been described in our previous study[20]. The monoclonal antibody mouse anti-TYLCV CP was kindly provided by Professor Xiaowei Wang (Institute of Insect Sciences, Zhejiang University). HRP-conjugated GST tag monoclonal antibody (HRP-66001) and HRP-conjugated His tag monoclonal antibody (HRP-66005) were purchased from Proteintech. Secondary goat anti-mouse IgG (ab6789) and goat anti-rabbit IgG (ab6721) antibodies were from Abcam.

**GST pull-down assays.** All pull-down assays were conducted by the GST Protein Interaction Pull-Down Kit (Pierce, 21516) according to the manufacturer's protocol. In brief, after IPTG induction, total proteins extracted from recombinant expression cells were incubated with the agarose for 1–2 h for GST-tagged proteins, followed by incubation with His tagged fusion proteins for 2–3 h. In competitive interaction assays, His-ATG8 and His-CP were simultaneously incubated with GST-PEBP4-bound agarose. The glutathione agarose samples that hardly be eluted by 10–50 mM glutathione was boiled within wash solution and SDS sampling buffer at 100 °C, and discarded after centrifugation (mainly His-ATG8 and His-Raf1). The supernatant was analyzed by Western blotting. For boiled samples, nonspecific bands could appear in controls due to nonspecific interaction of input samples with glutathione agarose.

**Co-immunoprecipitation (Co-IP) and in vitro protein co-incubation assays.** Co-IP experiments were conducted by the commercial Co-IP Kit (Pierce, 26149) according to the manufacturer's protocol. Total proteins extracted from whiteflies in the IP lysis buffer were subjected to Co-IP and co-incubation assays. The final elute of Co-IP was analyzed with Lane Marker Sample Buffer containing 100 mM DTT in a reducing gel. For protein co-incubation assays in vitro, whitefly lysate was directly incubated with the fusion protein for 2–3 h. The precipitation was discarded after centrifugation, and the supernatant was analyzed by western blotting.

**Immunofluorescence detection and TUNEL assays.** Collected whiteflies were fixed in 4% paraformaldehyde overnight and rinsed in TBST (TBS with 0.2% Triton X-100) for 30 min. TUNEL assays were carried out by In Situ Cell Death Detection Kit, POD (Roche, 11684817910) according to the manufacturer's protocol. For immunofluorescence, samples were blocked with Pierce SuperBlock™ T20, and incubated with the primary antibody at room temperature for 3 h. Following rinsing for five times in TBST (TBS with 0.05% Tween 20), samples were incubated with the secondary antibody at room temperature for 2 h. If two primary antibodies from the same host species were necessary, such as anti-Raf1, anti-PEBP4, and anti-ATG8, labeling was sequential. Two ATG8 antibodies were used in confocal assays: anti-ATG8/GABARAP (CST, 13733) was used to label ATG8-PE to determine autophagy intensity, and anti-GABARAP (Proteintech, 18723-1-AP) was used to label both ATG8 types to determine ATG8-PEBP4 interaction. The anti-mouse IgG conjugate Alexa 488 (ab150113), anti-rabbit IgG conjugate Alexa 555 (ab150078), anti-rabbit IgG conjugate Alexa 647 (ab150079), and anti-rabbit IgG F(ab) fragment conjugated Texas Red (ab7052) were purchased from Abcam and CST. Whitefly midguts and salivary glands were dissected after staining and mounted in the Fluoroshield Mounting Medium with DAPI (Abcam) before imaging under Zeiss LSM710 confocal microscope. Relative intensity of TUNEL and ATG8-PE was analyzed using ImageJ software.

**Pharmacological experiments and RNA interference via feeding.** Collected insects were placed in cylindrical, light-proof containers, 30 mm diameter × 60 mm height. Each container was provided with a 500 μl diet containing 15% final concentration of sucrose. MEK phosphorylation inhibitor mirdametinib (PD0325901) (Selleck, S1036), apoptosis inducer PAC-1 (Selleck, S2738), inhibitor Z-VAD-FMK (Selleck, S7023), and autophagy inducer rapamycin (Millipore, 553210) were dissolved in dimethyl sulfoxide (DMSO) and mixed in diet at a concentration of 10 μM. Autophagy inhibitor 3-MA (Selleck, S2767) was dissolved in double distilled water and incorporated into diet at 1 μM. Concentrations of agonists and inhibitors were based on previous studies[26,68]. An equivalent amount of DMSO was used as a feeding control. All chemicals were administered via 24 h feeding.

The T7 RiboMAX Express RNAi System (Promega, P1700) was used to synthesize double stranded RNA (dsRNA) according to manufacturer's protocol. dsRNAs, i.e. ds*PEBP4*, ds*Iap*, ds*Caspase3*, ds*mTOR*, ds*ATG8* (50 μl, 10 μg/μl) (dsGFP as control) were added to 450 μl artificial diet to feed insects. Their interference efficiencies were measured by RT-qPCR. All insects in the same container were collected as an independent biological replicate, and each treatment contained at least three replicates. To collect viruliferous whitefly, nonviruliferous whitefly was transferred to TYLCV-infected tomato plant for virus acquisition.

**Survival rate analysis.** Whiteflies, 100 per replicate, were fed in a feeding container for 24 h, then transferred to caged plant leaves for 48 h for TYLCV acquisition. Surviving whiteflies in each leaf-cage were recorded. To maximally activate autophagy, a time-course feeding treatment was performed that lasted for 72 h. At each time point, numbers of survivors in each of the three biological replicates were recorded. To determine the effect of virus overloading whiteflies were first fed on 3-MA for 24 h, and 60 individuals per biological replicate, were transferred to caged TYLCV-infected vs. uninfected plants for another 48 h. Three replicates in total were carried out.

**Statistical analysis and reproducibility.** Statistical analyses were performed with SPSS software (Chicago, IL, USA) and GraphPad software. All data were checked for normality by the Wilk–Shapiro test. Two-tailed paired *t*-test was used to separate the means of normally distributed data, while Mann–Whitney test was used to analyze nonparametric distributed data. Quadratic survival model was used to analyze the prediction bands (95% confidential interval) of whitefly survival rate data. All western blots and confocal assays were repeated independently at least three times with similar results.

**Reporting summary.** Further information on research design is available in the Nature Research Reporting Summary linked to this article.

## Data availability

All data are available within the article, supplementary information or source data file. All protocols have been described in Methods or in the references therein. Sequence information was acquired from NCBI and GIGA database. LIR motif was analyzed in iLIR Autophagy Database. Plasmids used in the study are freely available upon request. No custom code or mathematical algorithm was used in this work. Plant cultivars, virus, and insect may be available upon request after publication and respective material transfer agreements are completed. Source data are provided with this paper.

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

## Acknowledgements
We appreciate material support from Professor Xiaowei Wang and Jianxiang Wu, Zhejiang University, Professor Xueping Zhou, Institute of Plant Protection, CAAS, and Professor Chuanyou Li, Institute of Genetics and Development Biology, CAS. This project was supported by the Strategic Priority Research Program of the Chinese Academy of Sciences (Grant No. XDPB16 to Y.S. and F.G.) and the National Key Research and Development Plan (2017YFD0200400 to F.G.).

## Author contributions
S.W. designing project, performing experiments, analyzing data and writing manuscript; H.G. and K.Z. manuscript writing; F.G. and Y.S. project designing and coordination, manuscript writing.

## Competing interests
The authors declare no competing interests.
