## [Peer Review File · Nature Communications]

PEBP balances apoptosis and autophagy in whitefly upon arbovirus infectionREVIEWER COMMENTS

Reviewer #1 (Remarks to the Author):

The authors did a very nice job exploring the interactions between PEBP4, apoptosis, and autophagy and how these interactions impact TYLCV. The experiments were well thought out utilizing a number of different techniques. Below are my specific comments:

The manuscript requires heavy editing for grammatical errors. Often “the” is either excluded or included at inappropriate times making the sentence structure awkward and hard to enjoy.

It is unclear what the authors mean by “viral loading” and overloading? Presumably this refers to virus titers or viral burden but is unclear.

Fig. 2D It seems like PEBP4 is expressed in almost all of the midgut and salivary gland cells and that cells infected with TYLCV naturally express CP; however, these images don’t necessarily show co-localization. To make this claim the authors would have to show sub-cellular localization and it seems the magnification is not high enough to draw these conclusions.

The authors state (Ln 174-75) that the translational abundance of PEBP4 increase upon virus infection specifically 12-24 hpi, but this is not clear upon examination of the western blot (Fig S2B). The ratio of PEBP4 to GAPDH be can quantified in ImageJ to provide a quantifiable number which statistics can be performed on.

Overall, the microscopy is good, but I think their conclusions could be strengthened by including quantifiable metrics to their microscopy observations throughout the manuscript.

Fig 5D. Why did the levels of PEBP4 not increase as the concentration of input GST-PEBP4 was added to the system?

Figure 6F legend: Not sure what “in terms of over-activation of autophagy” means. The authors just measured survivorship. I think that statement can be removed.

The model figures presented in Figures 4 and 5 were really nice and informative, but in my opinion, Figure 6H could be improved to make it more visually appealing and informative. I found the current figure to be confusing.

Reviewer #2 (Remarks to the Author):

PEBP regulates balance between apoptosis and autophagy, enabling coexistence of arbovirus and insect vector.

This is an important and well-executed manuscript. It combines genetic and biochemical approaches in vitro and in vivo to provide novel insights into how cell death pathways influence virus infection in an insect host, balancing fitness of virus (tomato yellow leaf curl virus – TYLCV) and insect (whitefly *Bemisia tabaci*). The results are novel and, in my opinion, of general interest. They provide relevant insights on the contribution of different cell death pathways and self-degradative process (autophagy) in host defense against virus, from the cellular level to the whole-organism level.

The manuscript claims that a balance between autophagy and apoptosis influences arbovirus dynamics in the insect *Bemisia tabaci*. They identify PEBP4 as a host protein that binds an arbovirus protein (CT) and modulates two forms of programmed cell death, with opposing outcomes in terms of virus load in the host. The identification of PEBP4 as a host protein that modulates antagonistic arbovirus loading processes in an insect is an important contribution of the manuscript. However, some of the underlying principles are still unclear to me based on the arguments below:

1 – Major point: Since apoptosis and autophagy modulate virus loads in different directions and are active at the same time in the midgut and salivary gland tissue, authors have to explain where these two processes are happening during infection with more detail (ex: patch of midgut cells instead of zoomed-out images). We have little to no information on the spatial-temporal events that occur following infection. All immunofluorescence and TUNEL assay images provide a tissue-level picture of apoptosis, autophagy, PEBP4 and CP (not virus itself), but images seem to be used to express signal intensities, but not spatial-temporal details. For example, is it possible for an infected midgut cell to become apoptotic while a neighbor midgut infected cell activates autophagy? How this would influence virus spread? Does CT reach the intracellular environment only through a new TYLCV infection or it can be passed laterally between infected cells? This would allow programmed cell death through apoptosis or autophagy to be differentially initiated by its influence on PEBP4 binding before virus infection, giving time advantage to the initiation of autophagy or apoptosis. Does autophagy occur exclusively in infected cells? Does

apoptosis occur exclusively in infected cells? This would help clarify how PEBP4 balances apoptosis and autophagy and optimize arbovirus abundance, as claimed by the authors.

2 – Major point: Pharmacological assays concluded that “virus loading was predominantly determined by autophagy rather than by apoptosis”. This approach fails to clarify how the balance between autophagy and apoptosis results in an equilibrium between virus loads and host fitness because it ignores the different drug pharmacokinetics of each tested chemical and how they influence the magnitudes of activation and inhibition of apoptosis and autophagy (Figure 6 – rapamycin, 3-MA, Z-VAD, PAC-1).

Other points:

3 – Most of the observed phenotypes seem to happen in the midgut and salivary glands exactly in the same manner, irrespective of viral dynamics and spread in the insect body. Authors should make clear the time post-infection of each experiment (in figure legend at least). I assume virus spread leads to the midgut and salivary gland being infected at different time points in vivo, so it's surprising that apoptosis and autophagic processes described here occur at the same time and intensity in both organs.

4 – Line 159 – Figure 2D suggests CP and PEBP4 colocalization in vivo is a minor event in the midgut. Can the author include an analysis of how representative is this interaction using several midgut replicates? This will inform how predominant in the midgut is the interaction between CP-PEBP4.

5 – Line 217. If the authors want to claim that “both PEBP4 and TYLCV CP directly increased the unphosphorylated Raf1, which equals to decrease phosphorylated Raf1 because no additional Raf1 products could be supplied in whitefly lysate, resulting in reducing the phosphorylation of Raf1” they need to show in the in vitro assay p-Raf1 signal decreasing. Without this, it's difficult to understand the increased appearance of Raf1 signal in figure 4D.

6 – Line 221-222. About phosphorylation of Raf1. Authors claim “Taken together, TYLCV infection not only increased unphosphorylated Raf1 at the transcriptional level but directly reduced the phosphorylation of Raf1 at the protein level.” Based on the data that is presented, I don't see this conclusion is supported by the experiments. It is clear that CT increases Raf1 at the protein levels. Nothing was done (as far as I understand) about its phosphorylation status.

7 – About figure 4:

7.1 – Figure 4D: Experiments 4D is tricky and it would look more convincing if authors could use a specific p-Raf1 antibody.

7.2 – Figure 4F: Please, indicate what antibody was used to immune-precipitate proteins for the Co-IP experiment shown.

7.3 – Figure 4I: Raf1 signal decrease in the presence of a MAPK inhibitor. Shouldn't it be unchanged since its phosphorylation status is what governs Raf1 action in this pathway?

7.4 – Since PEBP4 interaction with CT and Raf1 occurs in different domains of PEBP4, it is not clear how and if the presence of CT influences Raf-1 phosphorylation, as proposed in the model presented in figure 4J. Specific antibodies that differentiate p-Raf1 from Raf1 would be a critical reagent.

7.5 – Line 224. Figure 4F. It is not clear to me which protein is immunoprecipitated. Is it PEBP4? Please, clarify. Besides, PEBP antibody signal is pretty faint in immunoblots shown in 4D and F, making it difficult to conclude its presence.

8 – Line 326. I'm not convinced about the conclusion of figure 6F. The simplest explanation is the more chemicals you add to your survival curve, the more you enhance insect mortality due to off-target toxic effects. How to exclude this possibility? How did the authors decide on the doses used? Did they perform preliminary dose-response experiments?

9 – Line 327. I had trouble understanding the study design of Figure 6G. Why is the ratio between control and infected plant has to be accounted for in order to test how autophagy suppression impacts the survival of virus-infected insects? Please, clarify.

10 – Line 443: "The homeostasis shaped by apoptosis and autophagy provides an immune tolerance, allowing TYLCV persistently preserved in whitefly population". I'm not opposed to the use of the word tolerance here. However, disease tolerance was not measured. As a dynamic trait, disease tolerance can only be measured as a dose-response curve relating host health (ex: survival) x microbe load. The tolerance measure is the slope of this curve (Simms. Defining tolerance as a norm of reaction. Evolutionary Ecology volume 14, pages563–570(2000).

11 – Line 27. Abstract. In the sentence, the authors mention a "mild immune response" after measuring parameters related to apoptosis and autophagy. No other immune parameter was evaluated, so the sentence, in my opinion, is inaccurate. Besides this, why "mild"? What has been comparatively measured to allow this adjective?

12 – Line 10-11. Check sentence to improve clarity. “Pharmacology assays show that apoptosis promotes TYLCV loading in whitefly depends its inhibition on anti-viral autophagy”.

13 – Line 217. Misspelling “de novo”.

14 – Line 224. “In florescence” is misspelled.

Jose Henrique M. Oliveira (<https://orcid.org/0000-0003-3814-5312>). Federal University of Santa Catarina. Brazil.

Reviewer #3 (Remarks to the Author):

This manuscript investigated the interplay between virus (Tomato yellow leaf curl virus – TYLCV) and the insect protein PEBP4 to regulate apoptotic and autophagy responses for maintaining the balance between the virus levels and the insect immune response and the effects that the virus may cause to insect.

Before going into the detailed review, I have to raise the fact that the level of English in this manuscript, at least at the beginning, needs substantial improvement, and I assume the authors didn't send it for language editing. Throughout the manuscript, I had difficulties following the results because of the English level.

Major comments:

- The results presented in figure 1 suggest that autophagy and apoptosis were activated 24 h after TYLCV acquisition. This is a very surprising result, as the virus is circulative and needs about 8 hours to be transmitted. I thus would wonder why the virus needs to activate apoptosis and autophagy if it is transmitted and secreted that fast outside the body. If the virus is also replicating, which is a very controversial issue for this specific virus, I expect to see the effects on autophagy and apoptosis after

more than 24 h, but again if the virus already outside the body, why it would need to activate such mechanisms? I would test additional time points and repeat the staining and gene expression results presented in these experiments to see whether the virus is retained for longer time periods for other purposes in the insect cell, for example for replication, which then needs to be shown. Furthermore, cleavage of caspase-3 was observed 6 hours following acquisition. This is a very short time for a virus that is not very well established as a replicating virus, and thus I would again test whether the virus enters the nuclei in the gut and the hemolymph and replicates to support the presented results.

- In the next set of experiments the authors identified PEBP4 as interacting with the virus CP. It is not mentioned which other proteins interacted with the virus CP from the whitefly library that was screened, or whether this was the only interacting protein? If other proteins were identified, were they investigated? And what is the rationale for choosing PEBP4 and not others? It is mentioned that it was investigated for its putative function in regulating PCD, although this group of proteins is a very large family of more than 200 proteins that have been found to be highly expanded in whiteflies, and is probably the largest in insects. They have diverse functions in the cell and interact with many cellular proteins.

- The quality of the staining images of PEBP4 and the virus in the gut and salivary glands is very poor and we don't really see any co-localizations, but overlap of the two colors. Zoom-in images should be provided. Immunostaining of virus particles usually results in even single viral particles or small aggregates which should be easily visible. The images provided show a staining which is highly nonspecific and not associated with any cellular compartments.

In the same set of experiments, the signals obtained in the pull down assay results are very weak and the bands in the second panel of the pull downs are weak and almost not seen, for example the GST-PEBP4 bands. I am not sure any bands can be seen. Those should be provided with better quality and convincing signals.

- The same problem with the microscopy results can be seen in figures 3D and E. The images are very small and it is very hard to see any staining details. For example, comparing silenced with non silenced organs does not show real difference in the staining and those signals can be quantified. Those images are qualitative and the TUNEL signals, that are comparable, can be seen in both with almost the same intensities. The same thing with the immunostaining results which are a mix of colors that do not conclude the claimed results. For example the salivary gland seen in the silenced samples is shown with three colors that one cannot determine any subcellular conclusions based on those image. If the intensities are meant to be show, they should be quantified because those images are not informative. The same imaging problems can also be seen in the next figures of the manuscript. It has to be noted that both the midgut and the salivary glands were always used to explain the observed interactions, however, subcellular images have never been shown.

- The interaction of PEBP4 with the virus CP and Raf1 which regulates the MAPK phosphorylation and enhance virus-induced apoptosis has been shown in other systems, and thus it is descriptive and a confirmation of the results that has been published in other systems. The same conclusion can be also drawn with the interaction of the virus CP with PEBP4 to affect autophagy, a result that has also been shown in this system, however, now with a more mechanistic description, but again, has been show to be active in human systems.

In conclusion, the results presented here, although were executed by a large number of experiments, they are all planned based on the same interactions known in other systems such as human and other virus-vector systems. The ideas presented are not unique to the system (whitefly-virus) studied, and do not present a discovery of a new and novel mechanism that is specific for whitefly-TYLCV interactions. It is known, and has been well studied, that pathogens, manipulate immunity responses in their hosts to ensure they survival on one hand, and ensure they transmission on the other hand, for those who are vector-borne. The fact that TYLCV is not in the consensus as a replicating virus inside its whitefly vector, and many experimental evidence has been published and demonstrated that it is not a replicating virus, makes it a must y the authors of this manuscript to demonstrate these lacking results, namely, replication, entry into the nuclei and better evidence of spatial interactions as the microscopy images presented here are poor. I would suggest using insect cell lines to demonstrate these interactions, the closest approach to an in vivo situation.

Minor comments:

- Line 17: Autophagy is not a form of cell death

- Line 18: replace its with their

- Line 22: no need for "the"

- Line 24: and eliminated not eliminates

- Lines 71-82 is a paragraph that is attempting to review the apoptotic and autophagy related proteins that have roles in the response to infection, including viruses. However, this paragraph gives a list of proteins, without explaining the connections between them, their roles and why they are important for this study.

Below is our point-by-point response to reviewers' comments. The criticisms are in RED, and our responses are in BLUE.

Section 1 - Editor Comments

Thank you again for submitting your manuscript "PEBP regulates balance between apoptosis and autophagy, enabling coexistence of arbovirus and insect vector" to Nature Communications. We have now received reports from 3 reviewers and, after careful consideration, we have decided to invite a major revision of the manuscript.

As you will see from the reports copied below, the reviewers raise important concerns. We find that these concerns limit the strength of the study, and therefore we ask you to address them with additional work. Without substantial revisions, we will be unlikely to send the paper back to review.

In particular, there are concerns about the proposed mechanism and all reviewers agree that there would need to be convincing data about PEBP4 subcellular localization, whether and how autophagy or apoptosis occurs in infected cells, proposed mechanism of viral infection and tissue tropism, as well as concerns about microscopy that must be addressed.

Part I - Concerns about immunofluorescence images

- To address the concern expressed by all reviewers regarding subcellular localization of PEBP4, additional immunofluorescence experiments were conducted. A new set of TYLCV CP and PEBP4 co-localization images in midgut and salivary glands of whitefly have been present in the current version of manuscript. Here stronger PEBP4 fluorescence (red) was facilitated by a secondary antibody (anti-rabbit Alexa Fluor 647) purchased from Abcam (**Figure 2D**). The confocal microscope equipped with 63X objective lens greatly improved the interaction details. **We found that PEBP4 signals were largely associated with the plasma membrane, and a small portion of PEBP4 signal existed in cytoplasm.** To further confirm this finding, Z-Stack function with spatial scanning was used to scan different layers of midgut and salivary gland to determine the stereo localization of PEBP4. Yellow spots (arrow marked) represented the interaction of virus CP (green) and PEBP4 (red). At the subcellular level, PEBP4 without virus infection was also detected on plasma membrane (Figure S2). The 3D models of Figure 2D were placed in Video S1. Furthermore, the subcellular location of PEBP4 was also confirmed by its interaction with Raf-1 that is largely localized on plasma membrane (Lavoie *et al.*, *Nat Rev Mol Cell Biol*, 2020), as well as with ATG8 that is localized on autophagosome membrane (Kabeya *et al.*, *J Cell Sci*, 2004). The relevant description has been modified in Line 184-188 of main text.

Lavoie *et al.*, *Nat Rev Mol Cell Biol*, 2020: <https://doi.org/10.1038/s41580-020-0255-7>

Kabeya *et al.*, *J Cell Sci*, 2004: <https://doi.org/10.1242/jcs.01131>

Figure 2D

Figure S2

These new images reflected the max. resolution of Zeiss LSM710 fluorescence microscope equipped with 63X oil lens. The CP-Raf1-PEBP4 triple interaction (white spots in enlarged images) has been shown in both midgut and salivary gland of viruliferous whitefly (Figure 4G), whereas natural Raf1-PEBP4 interaction without virus infection (arrow-pointed purple spots) has also been shown in Figure S4C. Notably, most of these interactions occurred in plasma membrane.

Figure 4G

Figure S4C Raf1 (blue) interacts with PEBP4 (red)

Furthermore, the ATG8 antibody we used in Figure 1 only labels activated ATG8, missing its interaction with PEBP4 (PEBP4 liberates ATG8 to activate ATG8 lipidation). Another ATG8 antibody was therefore used to label all ATG8. Figure S5B shows that PEBP4 interacted with ATG8 in membrane. Moreover, the PEBP4-bound ATG8 should be inactivated because activated ATG8 is mostly located in autophagosome membrane (Kabeya *et al.*, *J Cell Sci*, 2004). Figure 5G showed that, after virus infection, PEBP4 interacted with ATG8 (purple spots) and virus CP (yellow spots) respectively, but no merged ATG8-CP cyan was found, indicating that CP competed with ATG8 for PEBP4-binding. Moreover, Figure 3E has been adjusted to keep the same fluorescence colour scheme with Figure 4G and Figure 5G. We hope these confocal results have provided sufficient details of PEBP4 subcellular localization and its interactions with Raf1 and ATG8.

Figure 5G

Figure S5B

Part II - Autophagy or apoptosis occurring in infected cells

- TYLCV has been found to invade multiple types of tissues including midgut, salivary gland, ovary, fat body, and even brain (Czosnek *et al.* 2017, Wang *et al.* 2020), and the occurrence of autophagy or apoptosis in viruliferous whitefly has been well established in a number of studies. See evidence we provide below.

Czosnek *et al.* 2017 <https://doi.org/10.3390/v9100273>

Wang *et al.* 2020 <https://doi.org/10.7554/eLife.56168>

Autophagy in TYLCV infect cells:

Reference: Wang *et al.* 2016 <https://doi.org/10.1080/15548627.2016.1192749>

However, they did not detect co-localization of CP and ATG8 in midgut cells, supporting our idea of ATG8 liberation rather than binding to CP-PEBP4 complex once CP hijacked PEBP4.

Wang *et al.* 2016

Apoptosis in TYLCV infect cells:

Reference: Wang *et al.* 2020 <https://doi.org/10.1128/mSystems.00433-20>

TYLCV infection also induces apoptosis in midgut and salivary gland in whitefly, reflected by caspase3 cleavage, the hallmark of apoptosis.

Wang *et al.* 2020

Reference: Wang *et al.* 2020 <https://doi.org/10.7554/eLife.56168>

In this article, apoptosis was induced in the brain of TYLCV infected whitefly, and caspase3 cleavage, the hallmark of apoptosis, was activated in infected cells.

Wang *et al.* 2020

Section 2 - Reviewer's Comments:

Reviewer #1:

The authors did a very nice job exploring the interactions between PEBP4, apoptosis, and autophagy and how these interactions impact TYLCV. The experiments were well thought out utilizing a number of different techniques. Below are my specific comments:

1. The manuscript requires heavy editing for grammatical errors. Often “the” is either excluded or included at inappropriate times making the sentence structure awkward and hard to enjoy.

➤ We have put significant effort to address this issue.

2. It is unclear what the authors mean by “viral loading” and overloading? Presumably this refers to virus titers or viral burden but is unclear.

➤ By virus loading or overloading, we attempted to emphasize the relation between virus abundance and vector capacity. In natural condition, virus titer may reach a high level during the acquisition period, but such a viral burden may not exceed vector's capacity. Conversely, if the vector's antiviral immunity is suppressed e.g. by pharmacological feeding, it could be overloaded by virus of relatively low titering. Modification has been made in the text (see Ln334-335).

3. Fig. 2D It seems like PEBP4 is expressed in almost all of the midgut and salivary gland cells and that cells infected with TYLCV naturally express CP; however, these images don't necessarily show co-localization. To make this claim the authors would have to show sub-cellular localization and it seems the magnification is not high enough to draw these conclusions.

➤ Our Z-stack showed that PEBP4 was localized in almost all midgut and salivary gland cells, and that CP-PEBP4 interaction (yellow spots) was mostly found in plasma membrane (in the middle and top layers of salivary gland pointed by arrows). Images from midgut also confirmed this result. The enlarged images displayed the interaction at subcellular level even though nuclei may not be shown because of the spatial location. The images of salivary gland and enlarged midgut reached the Max. Resolution of Zeiss LSM710 equipped with 63X oil lens. Yellow spots in the enlarged images represented merged signals of CP (green) and PEBP4 (red).

Figure 2D

PEBP4 and CP are abundant in infected cells, and both are able to bind to multiple proteins. 1) Y2H results (Table 1) showed that CP may interact with at least 23 putative targets, and some of which have been identified previously (Wei *et al.*, 2017; Kanakala and Ghanim, 2016; Gorovits *et al.*, 2013). 2) PEBP family was reported to regulate multiple signalling pathways by targeting several proteins or by being phosphorylated at different sites (Noh *et al.*, 2016; Schoentgen and Jonic, 2018; Yeung *et al.*, 1999; Garcia *et al.*, 2009). 3) Y2H result in Figure 2A showed CP-PEBP4 interaction was not strong (positive in TDO but negative in QDO). It was unlikely that this interaction led to direct protection/degradation of virus. Instead, it acted as a molecular regulator of key signalling pathways of intracellular immune to combat with virus, so the proportion of the interaction events may not be an important issue.

Wei *et al.*, *PNAS*, 2017: <https://doi.org/10.1073/pnas.1701720114>

Kanakala and Ghanim, *Front. Plant Sci.*, 2016: <https://doi.org/10.3389/fpls.2016.01702>

Gorovits *et al.*, *plos one*, 2013: <https://doi.org/10.1371/journal.pone.0070280>

Noh *et al.*, *Autophagy*, 2016: <https://doi.org/10.1080/15548627.2016.1219013>

Schoentgen and Jonic, *arXiv*, 2018: <https://arxiv.org/abs/1802.02378>

Yeung *et al.*, *Nature*, 1999: <https://doi.org/10.1038/43686>

Garcia *et al.*, *EMBO Reports*, 2009: <https://doi.org/10.1038/embor.2009.4>

4. The authors state (Ln 174-75) that the translational abundance of PEBP4 increase upon virus infection specifically 12-24 hpi, but this is not clear upon examination of the western blot (Fig S2B).

The ratio of PEBP4 to GAPDH be can quantified in ImageJ to provide a quantifiable number which statistics can be performed on.

- As suggested, ratios of PEBP4 to GAPDH from six biological replicates were used for statistical analysis. After an initial decrease upon TYLCV acquisition (0-6 hpi), abundance of PEBP4 maintained at a constant level from 6 to 48 hpi (Figure S3A). However, the *PEBP4* transcript increased after 48 hpi (Figure S3B), suggesting the newly produced PEBP4 were continuously consumed by TYLCV. We modified the text accordingly in Line 203-206.

Supplementary Figure 2

5. Overall, the microscopy is good, but I think their conclusions could be strengthened by including quantifiable metrics to their microscopy observations throughout the manuscript.

- ImageJ was used to quantify the relative intensities of TUNEL and ATG8 (Figs, 1A, 1B, 3D, 3E, 4H).

Interactions in other confocal assays were not quantifiable since PEBP4 (largely localized in cell plasma membrane) and DAPI staining (the nucleus reference) could not be captured

simultaneously. We have provided new images with different layers of tested tissues in this revision.

6. Fig 5D. Why did the levels of PEBP4 not increase as the concentration of input GST-PEBP4 was added to the system?

- Shown in the PEBP4 blot in Figure 5D reflects the endogenous PEBP4 level in total lysate. It was not affected by increasing GST-PEBP4 fusion protein (input). ATG8 had no change either. In contrast, in Figure 5E, although endogenous PEBP4 showed no change, ATG8 activation was suppressed by GST-PEBP4 because autophagy had already been activated by virus CP. Together, these results showed that PEBP4 is necessary for CP activating autophagy. We added a relevant statement in the manuscript (Line 278-280).

7. Figure 6F legend: Not sure what “in terms of over-activation of autophagy” means. The authors just measured survivorship. I think that statement can be removed.

- The statement has been removed as suggested. The accumulative survival rate curve (Figure 6F) was further analysed by quadratic survival model (regression model). Prediction band (95% CI) indicated the tendency of survivorship (Line 330-333). Additional RNAi assays were conducted to further support our conclusion that a persistent autophagy impaired whitefly survivorship (Figure S6H)

8. The model figures presented in Figures 4 and 5 were really nice and informative, but in my opinion, Figure 6H could be improved to make it more visually appealing and informative. I found the current figure to be confusing.

- We designed a new Figure 6H.

Figure 6H

Reviewer #2:

This is an important and well-executed manuscript. It combines genetic and biochemical approaches in vitro and in vivo to provide novel insights into how cell death pathways influence virus infection in an insect host, balancing fitness of virus (tomato yellow leaf curl virus – TYLCV) and insect (whitefly *Bemisia tabaci*). The results are novel and, in my opinion, of general interest. They provide relevant insights on the contribution of different cell death pathways and self-degradative process (autophagy) in host defense against virus, from the cellular level to the whole-organism level.

The manuscript claims that a balance between autophagy and apoptosis influences arbovirus dynamics in the insect *Bemisia tabaci*. They identify PEBP4 as a host protein that binds an arbovirus protein (CT) and modulates two forms of programmed cell death, with opposing outcomes in terms of virus load in the host. The identification of PEBP4 as a host protein that modulates antagonistic arbovirus loading processes in an insect is an important contribution of the manuscript. However, some of the underlying principles are still unclear to me based on the arguments below:

1 – Major point: Since apoptosis and autophagy modulate virus loads in different directions and are active at the same time in the midgut and salivary gland tissue, authors have to explain where these two processes are happening during infection with more detail (ex: patch of midgut cells instead of zoomed-out images). We have little to no information on the spatial-temporal events that occur following infection. All immunofluorescence and TUNEL assay images provide a tissue-level picture of

apoptosis, autophagy, PEBP4 and CP (not virus itself), but images seem to be used to express signal intensities, but not spatial-temporal details. For example, is it possible for an infected midgut cell to become apoptotic while a neighbor midgut infected cell activates autophagy? How this would influence virus spread? Does CT reach the intracellular environment only through a new TYLCV infection or it can be passed laterally between infected cells? This would allow programmed cell death through apoptosis or autophagy to be differentially initiated by its influence on PEBP4 binding before virus infection, giving time advantage to the initiation of autophagy or apoptosis. Does autophagy occur exclusively in infected cells? Does apoptosis occur exclusively in infected cells? This would help clarify how PEBP4 balances apoptosis and autophagy and optimize arbovirus abundance, as claimed by the authors.

Here are our point-by-point responses:

1-1 Authors have to explain where these two processes are happening during infection with more detail (ex: patch of midgut cells instead of zoomed-out images).

- TYLCV-induced apoptosis and autophagy have been previously reported in TYLCV-infected whitefly midgut and salivary gland cells (Wang *et al.* 2016, Wang *et al.* 2020). In our study, PEBP4 was identified as a regulatory node that participated in apoptosis and autophagy activations during TYLCV infection. Our new confocal data showed that PEBP4 largely localized in cell plasma membrane. Most of PEBP4-CP interaction were found on plasma membrane (Figure 2D).

Figure 2D

We found these three signals of CP, PEBP4, and Raf1 were merged close to the membrane (Figure 4G), supporting our previous finding on this triple interaction in Figure 4F.

Figure 4G

Moreover, CP activated autophagy by breaking PEBP4-ATG8 association rather than directly binding to ATG8. From our new confocal images, we found separate binding of PEBP4 to CP and ATG8 in the membrane, but not merged signals of CP-ATG8 (Figure 5G). With the new antibody, activated (free) and inactivated (associated with PEBP4) ATG8 could be separated, and the PEBP4-ATG8 interaction in plasma membrane was found in nonviruliferous whitefly (Figure S5B). Considering the localization of ATG8 that reported by previous study (Kabeya *et al.*, 2004), we assume that inactivated ATG8 may bind to PEBP4 in plasma membrane, and the activated ATG8 could be released to cytoplasm to initiate autophagosome formation. All interactions at the subcellular level have been shown in enlarged images.

Wang *et al.*, 2016 <https://doi.org/10.1080/15548627.2016.1192749>

Wang *et al.*, 2020 <https://doi.org/10.1128/mSystems.00433-20>

Kabeya *et al.*, 2004: <https://doi.org/10.1242/jcs.01131>

Figure 5G

Figure S5B

1-2 Is it possible for an infected midgut cell to become apoptotic while a neighbor midgut infected cell activates autophagy? How this would influence virus spread?

- In terms of process, autophagy and apoptosis often occur in the same cells (Maiuri *et al.*, *Nature Reviews*, 2007). In our study, PEBP4-CP simultaneously activated apoptosis and autophagy. Domain prediction of PEBP4 sequence suggest it is not an exocrine protein. Hence, we suggest PEBP4-balanced apoptosis and autophagy occur within the same cells.

Mariño *et al.*, 2014

In terms of outcome, your scenario is possible. Based on previous publications, the outcome of each infected cell could be determined by stress it confronted (Mariño *et al.*, *Nature Reviews*, 2014; Maiuri *et al.*, *Nature Reviews*, 2007). Autophagy enables infected cells containing less virus to eliminate the invaders and promote cell survival. Infected cells with abundant virus, once the degradative capacity of lysosomes is saturated along with autophagosome accumulation, may facilitate permeability transition pore (PTP) opening and the lethal mitochondrial outer membrane permeabilization (MOMP) leads to cell death (Ma *et al.*, *Circulation*, 2012). Hence, the infected cell may have two different outcomes depending on virus load, or perhaps in between coordinated by the vector capacity. PEBP4 function should not be impacted by outcome, because it activates apoptosis and autophagy at the early stage of infection.

Mariño *et al.*, *Nature Reviews*, 2014: <https://doi.org/10.1038/nrm3735>

Maiuri *et al.*, *Nature Reviews*, 2007: <https://doi.org/10.1038/nrm2239>

Ma *et al.*, *Circulation*, 2012: <https://doi.org/10.1161/CIRCULATIONAHA.111.041814>

1-3 Does CT reach the intracellular environment only through a new TYLCV infection or it can be passed laterally between infected cells?

- We believe the reviewer meant to say “CP” and not ‘CT’? If so, CP can reach the intracellular environment only through a new TYLCV infection. Its N-terminus contains a nuclear localization signal (NLS) domain, and TYLCV CP is considered as a karyophilic protein (Kunik, *et al.*, *The Plant Journal*, 1998). In phloem associated cells of host plants, CP is localized in the cytoplasm at the early stage of infection, and in cytoplasm as well as nuclei at the late stage of infection (Gorovits, *et al.*, *Virus Research*, 2013). Even if CP is translated in host cell, it is located at nucleolus and nucleoplasm (Wang, *et al.*, *Frontiers in Plant Science*, 2017). No evidence has been shown that CP can move laterally between cells in both host plants and vector insects.

Kunik, *et al.*, 1998: <https://doi.org/10.1046/j.1365-3113X.1998.00037.x>

Gorovits, *et al.*, 2013: <https://doi.org/10.1016/j.virusres.2012.09.017>

Wang, *et al.*, 2017: <https://doi.org/10.3389/fpls.2017.02165>

1-4 Does autophagy occur exclusively in infected cells? Does apoptosis occur exclusively in infected cells?

- Apoptosis and autophagy are conserved cellular processes. We show here that the virus CP can directly activate apoptosis and autophagy via hijacking PEBP4. In fact, apoptotic and autophagic cells exist in midgut or salivary gland even without noticeable virus accumulation (Wang *et al.*, 2016, Wang *et al.*, 2020, and our results in this manuscript).

Wang *et al.* *Autophagy*, 2016 <https://doi.org/10.1080/15548627.2016.1192749>

Wang *et al.* *mSystem*, 2020 <https://doi.org/10.1128/mSystems.00433-20>

Although the mechanism of TYLCV activating autophagy has not been determined, an inflammatory signal cascade has been identified to be involved in apoptotic signal amplification in TYLCV-infected whitefly: the extracellular signal produced from infected cells could lead to apoptosis of uninfected cells (Wang *et al.*, 2020.). In our study, knockdown of PEBP4 significantly decreased CP abundance (Figure 3F), indicating that PEBP4-dependent activations on apoptosis and autophagy were required for virus loading in whitefly. Thus, although apoptosis and autophagy may occur in uninfected cells, process in TYLCV-infected cells is the most important determinant in virus loading in whitefly.

Wang *et al.* 2020 <https://doi.org/10.7554/eLife.56168>

2 – Major point: Pharmacological assays concluded that “virus loading was predominantly determined by autophagy rather than by apoptosis”. This approach fails to clarify how the balance between autophagy and apoptosis results in an equilibrium between virus loads and host fitness because it ignores the different drug pharmacokinetics of each tested chemical and how they influence the magnitudes of activation and inhibition of apoptosis and autophagy (Figure 6 – rapamycin, 3-MA, Z-VAD, PAC-1).

- Concentrations of these drugs fed to whitefly were previously reported, and were effective in activating or inhibiting apoptosis/autophagy (Wang *et al.* 2016, Wang *et al.* 2020). It is possible that simultaneously administering two drugs may complicate the outcome. To address this, a new set of RNAi experiments were conducted to confirm the pharmacological results (Figure S6). Because knockdown of Casp3, Iap, mTOR and ATG8 could either repress or derepress the process of apoptosis/autophagy, these target genes play a role in regulating

apoptosis/autophagy. Although the effective magnitudes could be different for RNAi and drugs, **similar tendencies of virus loading on whitefly survivorship were observed** (Figure S6E-H), supporting our main finding shown in Figure 6.

Results of oral delivery of drugs or dsRNAs indicated that the relative virus abundance was positively regulated by apoptosis but negatively regulated by autophagy (Figure 6A, 6B). Once autophagy was activated/inhibited, however, promoting effect via apoptosis on virus abundance diminished (Figure 6C, 6D, S6E, S6G), suggesting that autophagy was necessary for the positive effect of apoptosis on TYLCV. CP-PEBP4 enables a simultaneous activation of apoptosis and autophagy, rather than a mutual inhibition in general (Mariño *et al.*, *Nature Reviews*, 2014). This helps avoid over-activation of one of the processes in the cell that likely leads to a fitness cost (Figure 6F, S6H). Dual activation of these two processes with opposite effects on virus, ensures maintenance of an optimal virus load (Figure 6G; Czosnek and Ghanim, 2012; Wang *et al.*, 2020).

Drug/dsRNA function in Figure S6

Rapamycin/dsmTOR: activation of autophagy

3-MA/dsATG8: inhibition of autophagy

PAC-1/dslap: activation of apoptosis

Z-VAD/dsCasp3: inhibition of apoptosis

Mariño *et al.*, 2014: <https://doi.org/10.1038/nrm3735>

Czosnek and Ghanim, 2012: [https://doi.org/10.1016/S2095-3119\(12\)60007-0](https://doi.org/10.1016/S2095-3119(12)60007-0)

Wang *et al.* 2020 <https://doi.org/10.7554/eLife.56168>

Other points:

3 – Most of the observed phenotypes seem to happen in the midgut and salivary glands exactly in the same manner, irrespective of viral dynamics and spread in the insect body. Authors should make clear the time post-infection of each experiment (in figure legend at least). I assume virus spread leads to the midgut and salivary gland being infected at different time points *in vivo*, so it's surprising that apoptosis and autophagic processes described here occur at the same time and intensity in both organs.

- We did not pursue the time course of apoptosis/autophagy activation. Typically, salivary gland of insect vector is the last organ invaded by arbovirus, and in the whitefly-TYLCV system, this takes only a few hours. Therefore, 24 h acquisition is surely enough for TYLCV invading midgut and salivary gland. This was also supported by our confocal results showing that apoptosis and autophagy in both organs were activated by virus (Figure 1A, 1B). However, we did not intend to reveal tissue-specific response at different time points. Variations among individuals in virus acquisition makes it difficult if not impossible, unlike the mixed samples in western blotting.

4 – Line 159 – Figure 2D suggests CP and PEBP4 colocalization *in vivo* is a minor event in the midgut. Can the author include an analysis of how representative is this interaction using several midgut replicates? This will inform how predominant in the midgut is the interaction between CP-PEBP4.

- We did 10 replicates for salivary gland and 11 replicates for midgut of viruliferous whitefly and chose one representative in the revised Figure 2D. We did not quantify the fluorescence intensity because PEBP4 was dominantly localized in plasma membrane. Signals scanned from different layers were incomparable. As you can see, most interactions were found in the top layer (plasma membrane), but few were found in the middle layer where nuclei are. Although PEBP4 and CP were abundant in infected cells, but CP-PEBP4 interaction is a minor event when compared to unmerged CP and PEBP4. Because both are able to bind with multiple targets proteins. 1) Even in this study, Y2H results (Table 1) showed that CP may interact with at least 23 putative targets, and many other targets have been identified previously (Wei *et al.*, 2017; Kanakala and Ghanim, 2016; Gorovits *et al.*, 2013). 2) PEBP family can regulate multiple signalling pathways by targeting with different proteins and by being phosphorylated at different sites (Noh *et al.*, 2016; Schoentgen and Jonic, 2018; Yeung *et al.*, 1999; Garcia *et al.*, 2009). 3) Y2H result in Figure 2A showed that CP-PEBP4 is not a strong interaction (positive in TDO but negative in QDO). It was unlikely that this interaction functioned a direct protection/degradation on virus. Instead, it acted as a molecular regulator of key signalling pathways of intracellular immune to combat with virus, so the proportion of the interaction events may not be an important issue during this process.

Wei *et al.*, *PNAS*, 2017: <https://doi.org/10.1073/pnas.1701720114>

Kanakala and Ghanim, *Front. Plant Sci.*, 2016: <https://doi.org/10.3389/fpls.2016.01702>

Gorovits *et al.*, *plos one*, 2013: <https://doi.org/10.1371/journal.pone.0070280>

Noh *et al.*, *Autophagy*, 2016: <https://doi.org/10.1080/15548627.2016.1219013>

Schoentgen and Jonic, *arXiv*, 2018: <https://arxiv.org/abs/1802.02378>

Yeung *et al.*, *Nature*, 1999: <https://doi.org/10.1038/43686>

Garcia *et al.*, *EMBO Reports*, 2009: <https://doi.org/10.1038/embor.2009.4>

Figure 2D

5 – Line 217. If the authors want to claim that “both PEBP4 and TYLCV CP directly increased the unphosphorylated Raf1, which equals to decrease phosphorylated Raf1 because no additional Raf1 products could be supplied in whitefly lysate, resulting in reducing the phosphorylation of Raf1” they need to show in the in vitro assay p-Raf1 signal decreasing. Without this, it’s difficult to understand the increased appearance of Raf1 signal in figure 4D.

- We agree. After testing two pRaf1 antibodies (purchased from BBI): anti-pSer338 (D155090) and anti-pSer621 (D151522), we chose pSer621 that does not have non-specific bands for western blots. As expected, pRaf1 was suppressed by TYLCV. Results has been incorporated in the text (Figure 4C, 4D, 4E and 4I).

6 – Line 221-222. About phosphorylation of Raf1. Authors claim “Taken together, TYLCV infection not only increased unphosphorylated Raf1 at the transcriptional level but directly reduced the phosphorylation of Raf1 at the protein level.” Based on the data that is presented, I don’t see this conclusion is supported by the experiments. It is clear that CT increases Raf1 at the protein levels. Nothing was done (as far as I understand) about its phosphorylation status.

- See our response to the question above. We have also modified the text (see Line 246-256).

7 – About figure 4:

7.1 – Figure 4D: Experiments 4D is tricky and it would look more convincing if authors could use a specific p-Raf1 antibody.

➤ Completed as suggested.

7.2 – Figure 4F: Please, indicate what antibody was used to immune-precipitate proteins for the Co-IP experiment shown.

➤ Done as requested.

7.3 – Figure 4I: Raf1 signal decrease in the presence of a MAPK inhibitor. Shouldn't it be unchanged since its phosphorylation status is what governs Raf1 action in this pathway?

➤ This MAPK inhibitor Mirdametinib specifically targets MEK. Theoretically, it inhibits MEK and ERK phosphorylation instead of Raf1. Still, it successfully suppressed MAPK pathway. However, considering these samples were collected from living insects with multiple types of cells, it is possible that some feedback regulations existed, such as MEK signalling involved hyperphosphorylation of all six phosphorylation sites inhibits Ras-Raf1 interaction rather than stimulating it in general cases (Dougherty *et al.*, *Mol Cell*, 2005). In our study, decreased pRaf1 in Figure 4I indicated that it could be affected by downstream cascades, because Raf1/pRaf1 is not the target of Mirdametinib. We added this in Line 269-273.

Mirdametinib (PD0325901, <https://www.selleckchem.com/products/PD-0325901.html>)

Dougherty *et al.*, *Mol Cell*, 2005: <https://doi.org/10.1016/j.molcel.2004.11.055>

7.4 – Since PEBP4 interaction with CT and Raf1 occurs in different domains of PEBP4, it is not clear how and if the presence of CT influences Raf-1 phosphorylation, as proposed in the model presented in figure 4J. Specific antibodies that differentiate p-Raf1 from Raf1 would be a critical reagent.

➤ Results from pRaf1 antibody has been included in revised manuscript. Please check our responses to Comment 5&6.

7.5 – Line 224. Figure 4F. It is not clear to me which protein is immunoprecipitated. Is it PEBP4? Please, clarify. Besides, PEBP antibody signal is pretty faint in immunoblots shown in 4D and F, making it difficult to conclude its presence.

➤ We have now made it clear that PEBP4 was immunoprecipitated in Figure 4F (see Line 227). We did several times however the image quality has not been well improved. We think the weak PEBP4 signal could be due to (1) the inferior valency of the customized antibody (PEBP4 antibody) relative to the commercial antibodies; (2) source of PEBP4 in Figure 4F co-IPs. PEBP4 from whitefly protein lysate was further diluted when co-incubated with large volumes of His-CP and His-Raf1, leading to a very low level of the concentration of endogenous PEBP4.

8 – Line 326. I'm not convinced about the conclusion of figure 6F. The simplest explanation is the more chemicals you add to your survival curve, the more you enhance insect mortality due to off-target toxic effects. How to exclude this possibility? How did the authors decide on the doses used? Did they perform preliminary dose-response experiments?

➤ As indicated in our responses to the Major point 2, doses used was based on previous application in the same system (Wang *et al.* 2016, Wang *et al.* 2020). To address the "Off-

target toxic effects” issue you mentioned, we performed a new set of RNAi assays (Figure S6H) and observed a similar tendency to what’s seen in Figure 6F.

Wang *et al. Autophagy*, 2016 <https://doi.org/10.1080/15548627.2016.1192749>

Wang *et al. mSystem*, 2020 <https://doi.org/10.1128/mSystems.00433-20>

9 – Line 327. I had trouble understanding the study design of Figure 6G. Why is the ratio between control and infected plant has to be accounted for in order to test how autophagy suppression impacts the survival of virus-infected insects? Please, clarify.

- Whiteflies were raised on cotton (non-host for TYLCV) but tested on tomato, so the whitefly survivorship could be complicated by host shift. Using the ratio can exclude this factor and correct the data variation and bias.

10 – Line 443: “The homeostasis shaped by apoptosis and autophagy provides an immune tolerance, allowing TYLCV persistently preserved in whitefly population”. I’m not opposed to the use of the word tolerance here. However, disease tolerance was not measured. As a dynamic trait, disease tolerance can only be measured as a dose-response curve relating host health (ex: survival) x microbe load. The tolerance measure is the slope of this curve (Simms. Defining tolerance as a norm of reaction. *Evolutionary Ecology* volume 14, pages563–570(2000).

- Thanks for sharing this important publication, we did not use the word tolerance as a parameter, and we agree that it should reflect the association between host health traits and microbe load. Actually, as our knowledge, the concept of host tolerance (the ability to limit the disease severity induced by a given pathogen burden) was firstly made by Clunies-Ross in 1932, then resurfaced by plant ecologists in the 1994 (Simms and Triplett, 1994). However, the immune tolerance is more likely to be described as a state rather than measured as a parameter. For instance, “the states of nonresponse in immunity are termed as immune tolerance” (Wu *et al.*, 2019), and “immunological tolerance describes a diverse range of host processes that prevent potentially harmful immune responses within that host” (Waldmann, 2014). In our pharmacological and RNAi experiments, it is impossible to precisely control the virus abundance by individual test, so the dose-response curve could not be performed, but whitefly survivor rate differed with different virus load (Figure 6E, 6G, S6F) exhibited the tolerance association between whitefly and TYLCV (still, was not measured as a parameter).

Wu *et al.*, *Autophagy and Immune Tolerance*, 2019: https://doi.org/10.1007/978-981-15-0602-4_28

Waldmann, *Reference Module in Biomedical Sciences*, 2014: <https://doi.org/10.1016/B978-0-12-801238-3.00116-1>

11 – Line 27. Abstract. In the sentence, the authors mention a “mild immune response” after measuring parameters related to apoptosis and autophagy. No other immune parameter was evaluated, so the sentence, in my opinion, is inaccurate. Besides this, why “mild”? What has been comparatively measured to allow this adjective?

- We have changed the phrase “mild immune response” to “mild intracellular immune response”. Apoptosis, autophagy, and RNA interference are main intracellular immune mechanisms against arbovirus (Page 157, Bartholomay L C, Michel K. Mosquito immunobiology: the intersection of vector health and vector competence. *Annual Review of*

Entomology, 2018). Since TYLCV is a DNA virus, apoptosis and autophagy are considered as primary intracellular immunity that coordinately regulates TYLCV load in cells. We also rewrote the third paragraph in Introduction (Line 88-99) to define “mild immune response”, avoiding misleading the readers.

“Host infection by arbovirus often is acute, but persistent arbovirus can coexist with vector for generations, especially those that can be transmitted via developing eggs. Tolerance is usually used by vector insects to reduce deleterious effects meanwhile minimizing fitness cost. It maintains an optimal arbovirus transmitted relationship between viral loads and vector survival. Hence, a mild immune response could be the basis for virus tolerance in insect vector, which maintains an optimal arbovirus transmitted relationship between viral loads and vector survival. For instance, mosquitoes with impaired immune response reduce chikungunya virus infection. However, overly activated immune system causes production and accumulation of large amount cytokines, oxidative stress, and NO, resulting in autoimmunity and neurodegeneration of the insect vector. Apoptosis and autophagy are the best studied intracellular immune responses in virus-infected vectors, but how they shape vector’s immune tolerance to arbovirus, and how arbovirus in turn manipulates immunity in the insect vector remain unclear. Understanding such interaction, however, is crucial in control of vector-borne diseases.”

In our experiments, we considered the immune response triggered by agonist as “strong” and suppressed by inhibitor as “weak”. Both situations impaired whitefly survivorship either by causing immune burden or by overloading virus (Figure 6E-G, S6F, S6H). By contrast, the natural responses induced by TYLCV described as “mild” status, which indicates an optimal immune status that maintain a balance between physiological cost and virus retention in viruliferous whitefly. Similar description has also been used in mosquito-arbovirus system: “Overall, the cost of infection is usually **modest** and context-dependent, and mosquitoes are considered tolerant to arbovirus infections” (Lambrechts and Scott, 2009; Goic *et al.*, 2016).

Goic *et al.*, *Nature Communications*, 2016: <https://doi.org/10.1038/ncomms12410>

Lambrechts and Scott, *Cell Host & Microbe*, <https://doi.org/10.1016/j.chom.2019.08.005>

12 – Line 10-11. Check sentence to improve clarity. “Pharmacology assays show that apoptosis promotes TYLCV loading in whitefly depends its inhibition on anti-viral autophagy”.

➤ This sentence has been removed.

13 – Line 217. Misspelling “de novo”.

➤ It has been deleted (see Line 256).

14 – Line 224. “In florescence” is misspelled.

➤ It has been corrected (see Line 262).

Reviewer #3:

This manuscript investigated the interplay between virus (Tomato yellow leaf curl virus – TYLCV) and the insect protein PEBP4 to regulate apoptotic and autophagy responses for maintaining the balance between the virus levels and the insect immune response and the effects that the virus may cause to insect.

Before going into the detailed review, I have to raise the fact that the level of English in this manuscript, at least at the beginning, needs substantial improvement, and I assume the authors didn't send it for language editing. Throughout the manuscript, I had difficulties following the results because of the English level.

Major comments:

- The results presented in figure 1 suggest that autophagy and apoptosis were activated 24 h after TYLCV acquisition. This is a very surprising result, as the virus is circulative and needs about 8 hours to be transmitted. I thus would wonder why the virus needs to activate apoptosis and autophagy if it is transmitted and secreted that fast outside the body. If the virus is also replicating, which is a very controversial issue for this specific virus, I expect to see the effects on autophagy and apoptosis after more than 24 h, but again if the virus already outside the body, why it would need to activate such mechanisms? I would test additional time points and repeat the staining and gene expression results presented in these experiments to see whether the virus is retained for longer time periods for other purposes in the insect cell, for example for replication, which then needs to be shown. Furthermore, cleavage of caspase-3 was observed 6 hours following acquisition. This is a very short time for a virus that is not very well established as a replicating virus, and thus I would again test whether the virus enters the nuclei in the gut and the hemolymph and replicates to support the presented results.

- TYLCV reaches the midgut and salivary gland of whitefly in 40 min and 4–7 h respectively, after the onset of feeding, and peaks after 2 days (Czosnek *et al.*, 2002; Czosnek *et al.*, 2017; Pakkianathan *et al.*, 2015). Hence, it is reasonable that apoptosis and autophagy were activated in whitefly by TYLCV 24 hours. Figure 1E showed that autophagy and apoptosis were still proceeding from 24 to 48 hpi. As for a longer period, we assume that if new infection occurs both responses could also be activated, but this rarely happens because the amount of TYLCV peaks in whitefly after 2-3 days. The viruliferous whitefly carries TYLCV for its whole life whether the virus is propagative, so dual responses keep the virus at certain level without obvious cost on fitness, promoting the co-existence of the virus and its vector for long term.

A recent study has uncovered some details on TYLCV replicated in salivary gland of whitefly (He *et al.*, 2020). However, we did not find any merged signals of virus CP and nuclei in our results. In addition, no evidence suggests that virus replication is necessary for apoptosis and autophagy activation. Our results showed that both apoptosis and autophagy were activated by CP.

Czosnek et al., 2002

Czosnek et al., *Annals of Applied Biology*, 2002: <https://doi.org/10.1111/j.1744-7348.2002.tb00175.x>

Czosnek et al., *Viruses*, 2017: <https://doi.org/10.3390/v9100273>

He et al., *PNAS*, 2020: <https://doi.org/10.1073/pnas.1820132117>

Pakkianathan et al., *Journal of Virology*, 2015: <https://doi.org/10.1128/JVI.00779-15>

- In the next set of experiments the authors identified PEBP4 as interacting with the virus CP. **It is not mentioned which other proteins interacted with the virus CP from the whitefly library that was screened, or whether this was the only interacting protein? If other proteins were identified, were they investigated? And what is the rationale for choosing PEBP4 and not others?** It is mentioned that it was investigated for its putative function in regulating PCD, although this group of proteins is a very large family of more than 200 proteins that have been found to be highly expanded in whiteflies, and is probably the largest in insects. They have diverse functions in the cell and interact with many cellular proteins.

- Twenty-three putative targets of CP were identified by yeast-two hybrid experiment (Table S1) for possible interaction with CP, but none of them have been studied yet. PEBP4 was chosen due to the fact that this protein family has been previously proven to regulate apoptosis and autophagy in mammals, directly connected with phenotype of our interest (Figure 1). Modification has been made in Line 167-169.

Since CP only interacts with PEBP4 pre-domain rather than the conserved PE-binding domain, this regulatory function is very likely specific to PEBP4. In addition, PEBP4 was

localized in membrane (Figure 2D and Figure S2), making it possible to interact with Raf1 and ATG8 (also localized in membrane).

- The quality of the staining images of PEBP4 and the virus in the gut and salivary glands is very poor and we don't really see any co-localization, but overlap of the two colors. Zoom-in images should be provided. Immunostaining of virus particles usually results in even single viral particles or small aggregates which should be easily visible. **The images provided show a staining which is highly nonspecific and not associated with any cellular compartments.**

➤ We have repeated the experiments and provided a new set of images in Figure 2D, S2, 4G, S4C, 5G, S5B, Video S1-3.

Figure 2D

Figure S2

Figure 4G

Figure S4C Raf1 (blue) interacts with PEBP4 (red)

Figure 5G

Figure S5B

In the same set of experiments, the signals obtained in the pull down assay results are very weak and the bands in the second panel of the pull downs are weak and almost not seen, for example the GST-PEBP4 bands. I am not sure any bands can be seen. Those should be provided with better quality and convincing signals.

- We apologize for the weak signals. The expression efficiency of GST-PEBP4 was lower than GST-PEBP4 pep1. GST-PEBP4 signals were faint because GST-pep1 signals were too strong. Relevant raw figure was appended below, red arrows marked GST-PEBP4. Considering it was a GST pulldown assay and the interaction is clear (His-CP), we think this may not affect our conclusion, so we did not repeat this experiment in this version because of limited time.

- The same problem with the microscopy results can be seen in figures 3D and E. The images are very small and it is very hard to see any staining details. For example, comparing silenced with non silenced organs does not show real difference in the staining and those signals can be quantified. Those images are qualitative and the TUNEL signals, that are comparable, can be seen in both with almost the same intensities. The same thing with the immunostaining results which are a mix of colors that do not conclude the claimed results. For example the salivary gland seen in the silenced samples is shown with three colors that one cannot determine any subcellular conclusions based on those image. If the intensities are meant to be shown, they should be quantified because those images are not informative. The same imaging problems can also be seen in the next figures of the manuscript. It has to be noted that both the midgut and the salivary glands were always used to explain the observed interactions, however, subcellular images have never been shown.

- For Figure 3D and 3E, since we have quantified autophagy and apoptosis at both transcriptional and translational levels, quantification on fluorescence may not be necessary. Similar analysis can also be found in other publications (Chen et al., *PLoS Pathogens*, 2017; Wang et al. *Autophagy*, 2016; Vaidyanathan and Scott, 2006; Al-Olayan et al., 2002). As your request, ImageJ was used to quantify the relative intensities of TUNEL and ATG8 (Figs, 1A, 1B, 3D, 3E, 4H), relevant results were appended below. For quality issues of confocal images, we re-did these experiments to improve the details, and these results were updated. Please check Figure 2D, S2, 4G, S4C, 5G, S5B, Video S1-3.

Fig 1A,1B

Fig 3D,3E

Fig 4H

Chen *et al.*, *PLoS Pathogens*, 2017: <https://doi.org/10.1371/journal.ppat.1006727>

Wang *et al.* *Autophagy*, 2016: <https://doi.org/10.1080/15548627.2016.1192749>

Vaidyanathan and Scott, *Apoptosis*, 2006: <https://doi.org/10.1007/s10495-006-8783-y>

Al-Olayan *et al.*, *International Journal for Parasitology*, 2002: [https://doi.org/10.1016/S0020-7519\(02\)00087-5](https://doi.org/10.1016/S0020-7519(02)00087-5)

- The interaction of PEBP4 with the virus CP and Raf1 which regulates the MAPK phosphorylation and enhance virus-induced apoptosis has been shown in other systems, and thus it is descriptive and a confirmation of the results that has been published in other systems. The same conclusion can be also drawn with the interaction of the virus CP with PEBP4 to affect autophagy, a result that has also been shown in this system, however, now with a more mechanistic description, but again, has been show to be active in human systems.

- To our knowledge, whitefly PEBP has not been characterized for its biological function. This manuscript is the first study reporting that a PEBP protein from insect vector can be hijacked by an arbovirus protein. Wang and colleagues found that TYLCV activated whitefly autophagy, and autophagy degraded virus CP and DNA (Wang *et al.*, *Autophagy*, 2016). However, they did not provide any results about how CP activated autophagy in whitefly, and how autophagy coordinately regulates TYLCV abundance with apoptosis. Although some of the PEBP functions have been shown in medical research previously, our study provided molecular details on how arbovirus co-existed with its insect vector without significantly impairing vector's fitness.

In conclusion, the results presented here, although were executed by a large number of experiments, they are all planned based on the same interactions known in other systems such as human and other virus-vector systems. The ideas presented are not unique to the system (whitefly-virus) studied, and do not present a discovery of a new and novel mechanism that is specific for whitefly-TYLCV interactions. It is known, and has been well studied, that pathogens, manipulate immunity responses in their hosts to ensure they survival on one hand, and ensure they transmission on the other hand, for those who are vector-borne. The fact that TYLCV is not in the consensus as a replicating virus inside its whitefly vector, and many experimental evidence has been published and demonstrated that it is

not a replicating virus, makes it a must y the authors of this manuscript to demonstrate these lacking results, namely, replication, entry into the nuclei and better evidence of spatial interactions

- Pathogens manipulating immune responses have been well documented in many pathogen-host systems, but few have been reported in insect vectors, especially in vector and plant virus systems. To our knowledge, this is the first report demonstrating that PEBP was hijacked by a virus, and based on our results (Figure 2, Figure S1), the CP-PEBP4 interaction is unique in MED whitefly.

We did not discuss virus replication because it does not affect our findings whether virus replication within whitefly or not, since it is already circulative and persistent in whitefly. In addition, a latest research (He *et al.*, *PNAS*, 2020) provided some details on TYLCV replication in salivary gland of whitefly. Please check our response to Major comment 1.

He *et al.*, *PNAS*, 2020: <https://doi.org/10.1073/pnas.1820132117>

as the microscopy images presented here are poor. I would suggest using insect cell lines to demonstrate these interactions, the closest approach to an *in vivo* situation.

- The localizations of Raf1 and ATG8 have been well documented in previous studies. In MAPK signalling, “closed” Raf1 binds with the membrane associated GTPase and Ras, and the plasma membrane is the predominant activation site of MAPK activation (Lavoie *et al.*, *Nat Rev Mol Cell Biol*, 2020). Also, we re-did these experiments as suggested by reviewers to improve quality of images in Figure 2D, S2, 4G, S4C, 5G, S5B, Video S1-3. Our results showed that PEBP4 was mostly located in plasma membrane (Figure S2), and it suppressed MAPK signalling via interacting with Raf1 (Figure 4), we therefore thought that the PEBP4-Raf1 interaction mostly occurred in plasma membrane. On the other hand, inactivated ATG8/LC3 normally perform as cytosolic form. Once autophagy is activated, ATG8 conjugates with phosphatidylethanolamine (PE) and moves to autophagosomal membrane (Kabeya *et al.*, *J Cell Sci*, 2004). Human PEBP1 has been found specifically binds with PE-unconjugated LC3 (ATG8 homologue) to prevent autophagy activation (Noh *et al.*, *Autophagy*, 2016). In this study, ATG8 antibody we used (CST, 13733) enabled distinction of PE-conjugated and free ATG8. The illustration provided by the manufacturer on the antibody webpage was attached below. It therefore suggested that interaction of PEBP4 with free ATG8 may not be shown in confocal assay. We expected that, because PEBP4 can arrest or liberate ATG8 to regulate activation of autophagy, the endogenous PEBP4-ATG8 complex occurred in plasma membrane (Figure 5).

Official image of this antibody

Confocal immunofluorescent analysis of A172 cells, untreated (left) and chloroquine-treated (50 μ M, overnight; right), using GABARAP (E1J4E) Rabbit mAb (green) and β -Actin (8H10D10) Mouse mAb #3700 (red).

<https://www.cellsignal.com/products/primary-antibodies/gabarap-e1j4e-rabbit-mab/13733>

Cellular localization of GFP-LC3 and PEBP1 (Noh *et al.*, *Autophagy*, 2016)

Lavoie *et al.*, *Nat Rev Mol Cell Biol*, 2020: <https://doi.org/10.1038/s41580-020-0255-7>

Kabeya *et al.*, *J Cell Sci*, 2004: <https://doi.org/10.1242/jcs.01131>

Noh *et al.*, *Autophagy*, 2016: <https://doi.org/10.1080/15548627.2016.1219013>

Minor comments:

- Line 17: Autophagy is not a form of cell death

- In most cases, autophagy promotes cells to survive, but autophagic cell death (type II cell death) also happens. Description of terming autophagy as a type of cell death can also be found in some review articles. For example, “Two types of PCD have been described during infection in mosquito hosts: apoptosis and autophagy.” (Lyric C. Bartholomay and Kristin

Michel, Mosquito Immunobiology: The Intersection of Vector Health and Vector Competence, Annual Review of Entomology, 2018. "It (autophagy) has sometimes been viewed as a separate modality of programmed cell death" (Mariño *et al.*, *Nature Reviews*, 2014).

- Line 18: replace its with their

Done. - Line 22: no need for "the"

➤ Done.

- Line 24: and eliminated not eliminates

➤ Done.

- Lines 71-82 is a paragraph that is attempting to review the apoptotic and autophagy related proteins that have roles in the response to infection, including viruses. However, this paragraph gives a list of proteins, without explaining the connections between them, their roles and why they are important for this study.

➤ We replaced this paragraph with immune tolerance (Line 88-99) in Introduction, but in response of your concern, we tried to review apoptotic-autophagy crosstalk here.

Antagonism between apoptosis and autophagy has been reported in most cases despite other scenarios are also shown, i.e., apoptosis preceding autophagy, autophagy activating apoptosis. Both apoptosis and autophagy can be regulated by common signal-transducing molecules, such as Bcl-2 family. This family comprises proapoptotic/effector proteins (Bax and Bak), antiapoptotic proteins (Bcl-2, Bcl-xL, Mcl1, Bcl-W and A1), and BH3-only proteins (Bim, Bad, Bmf, Bid and PUMA). Together, these molecules maintain mitochondrial outer membrane permeabilization (MOMP) and endoplasmic reticulum (ER) calcium. The cell fate is determined by the ratio of Bcl-2 family members. All the subcellular localization, phosphorylation status, and the affinities of protein–protein interactions would break the homeostasis, and lead to different outcome. For instance, several BH3-only proteins directly interact with other multidomain proteins from the Bcl-2 family, thereby neutralizing those that have anti-apoptotic roles and stimulating those with pro-apoptotic functions to induce apoptosis. BH3-only proteins also competitively disrupt the inhibitory interactions between Beclin 1, one of the central regulators of autophagy, and the antiapoptotic proteins from the Bcl-2 family, then promote autophagy. The ratio of Bcl-2 family members is mediated by multiple signal transduction pathways, such as MAPK pathway. ERK activation typically inhibits apoptosis, and decrease on antiapoptotic proteins also liberates Beclin 1 to promote autophagy.

Mariño *et al.*, *Nature Reviews*, 2014: <https://doi.org/10.1038/nrm3735>

Maiuri *et al.*, *Nature Reviews*, 2007: <https://doi.org/10.1038/nrm2239>

REVIEWER COMMENTS

Reviewer #1 (Remarks to the Author):

The authors did a very thorough and nice job addressing the reviewers concerns. The amount of work and detail presented is impressive. While the infrequency of interactions captured by microscopy in many of the figures is reason for pause, the authors have provided supporting biochemical assays that provide confidence that the interactions are meaningful. I only have one major comment and the rest are minor.

Major:

I have been studying arbovirology for 20 years and this is the first I have seen the terms viral loading or overloading used. This terminology is confusing and detracts from the overall quality of the science. The authors should remove these terms from the manuscript and replace with common terms such as viral load, viral burden or viral titers.

Minor:

1. Grammatical and english syntax improvements are still need for the abstract and introduction.
2. Ln 14 Abstract Remove “most” or add “the”
3. Ln 37 “causes cells” seems like it was accidently left in the sentence and should be removed.
4. Ln 71-73 Not sure where this line or refence are coming from. This reference has nothing to do with invertebrate vectors.
5. Ln 70-71 Reference 30 does not demonstrate what is referenced in the sentence. In this study inhibiting the formation of viral-derived DNA in the vector significantly increased viral burden.
6. Ln 67-70 The two sentences are redundant and share very similar wordings.
- 7.

Reviewer #2 (Remarks to the Author):

I recommend the publication of the manuscript. Authors have made a significant effort to answer reviewer comments, especially with the microscopy and RNAi round of experiments. All my major concerns have been addressed.

Reviewer #4 (Remarks to the Author):

The authors have addressed most of the previous comments, except in the places where the authors kept on referring to published images, especially from Wang et al.2020.

Although the work is interesting, there is a major issue with this manuscript. Most of the confocal images used in this manuscript are not clear and are with very poor resolution. The 3D videos are not useful at all. The authors must be able to see the difference in their confocal images and in the published images which the authors kept mentioning while addressing the reviewer comments.

Line 94. Change to Whiteflies feeding on.

Fig. 1 legend. Time course immunoblot of guts or salivary glands? Why aren't the midguts photographed at 60x? The resolution would have been better and the signals would have been more prominent if the pictures were zoomed in.

Line 139. Why ' was further verified using Y2H'? It's already been selected from the Y2H assays.

Line 140. If there is no interaction in the stringent media (QDO), isn't the interaction weak?

Fig. 2A The growth is weaker even in TDO. Did the authors use X-gal to determine the interaction? If not, why?

Fig. 2D The signals for both PEBP and CP are weak. The colocalization in the midgut merged panel is weak but no doubt, shows colocalization, however, in the salivary glands, what the authors are claiming to be colocalization (yellow dots), seem to be an artifact. I do not understand why the pictures are not clear. I would also suggest the authors include the colocalization using ImageJ or any other available software. If already done, please put them in the figures or as sup fig for better understanding.

Also, put the intensity mapping in the sup. This will help the readers.

Line 148. How do the authors claim that PEBP was abundant on the plasma membrane? Did they use any plasma membrane tracker? The claim is not rational.

The videos of the immunolocalization aren't necessary. They don't give any better understanding. I suggest to remove them.

Fig. 3E. The signals would be better if taken at 60x for the midguts. The signals for Atg8-PE should also be analyzed using ImageJ for differences in intensity

Fig. 2B. From the western blots, it's visible that expression of PEBP is reduced in presence of TYLCV.

The authors should discuss this with proper reasoning.

Line 167. Why does the authors write that PEBP is 'consumed' by TYLCV? It implies TYLCV is not just binding to PEBP but is also breaking down the protein. Binding of TYLCV –CP to PEBP cannot decrease

the amount of protein. There must be some transcriptional repression / gene regulation behind it. The authors must discuss this in a proper manner.

Apoptosis and autophagy pathways are known to regulate each other. Authors mention that PEBP acts as a molecular switch between apoptosis and autophagy, and on the other hand the whole manuscript is based on activation of both apoptosis and autophagy together. As described in Fig. 6, I suggest the authors discuss more about the crosstalk in general and how it is different here where both are activated simultaneously.

Below is our point-by-point response to reviewers' comments. The criticisms are in RED, and our responses are in BLUE.

Reviewer #1 (Remarks to the Author):

The authors did a very thorough and nice job addressing the reviewers concerns. The amount of work and detail presented is impressive. While the infrequency of interactions captured by microscopy in many of the figures is reason for pause, the authors have provided supporting biochemical assays that provide confidence that the interactions are meaningful. I only have one major comment and the rest are minor.

Major:

I have been studying arbovirology for 20 years and this is the first I have seen the terms viral loading or overloading used. This terminology is confusing and detracts from the overall quality of the science. The authors should remove these terms from the manuscript and replace with common terms such as viral load, viral burden or viral titers.

- We have changed “viral loading” and “overloading” to “viral load” or “virus burden” in the main text as suggested. Thanks.

Minor:

1. Grammatical and english syntax improvements are still need for the abstract and introduction.
 - We carefully revised Abstract and Introduction to address the concern.
2. Ln 14 Abstract Remove “most” or add “the”
 - Removed
3. Ln 37 “causes cells” seems like it was accidently left in the sentence and should be removed.
 - It has been removed.
4. Ln 71-73 Not sure where this line or refence are coming from. This reference has nothing to do with invertebrate vectors.
 - We removed the phrase “insect vector”.
5. Ln 70-71 Reference 30 does not demonstrate what is referenced in the sentence. In this study inhibiting the formation of viral-derived DNA in the vector significantly increased viral burden.
 - It should be DENV. We rewrote this sentence and replaced the previous reference with an appropriate case study.
“Silencing the initiator caspase Aedronc reduces DENV-2 dissemination and infectious titer in mosquitoes.”
M. W. Eng, M. N. van Zuylen, D. W. Severson, Apoptosis-related genes control autophagy and influence DENV-2 infection in the mosquito vector, *Aedes aegypti*. *Insect Biochem Mol Biol* 76, 70-83 (2016).
6. Ln 67-70 The two sentences are redundant and share very similar wordings.
 - We deleted the redundant sentence. Thanks.

Reviewer #2 (Remarks to the Author):

I recommend the publication of the manuscript. Authors have made a significant effort to answer reviewer comments, especially with the microscopy and RNAi round of experiments. All my major concerns have been addressed.

Reviewer #4 (Remarks to the Author):

The authors have addressed most of the previous comments, except in the places where the authors kept on referring to published images, especially from Wang et al.2020.

- Actually, two different studies in citations were abbreviated with Wang et al. 2020. We listed these two citations, and highlighted study below was conducted by our lab, which is also a basis and motivation for this study.

S. Wang, et al., Apoptotic neurodegeneration in whitefly promotes the spread of TYLCV. *eLife* 9, e56168 (2020).

X. R. Wang et al., Apoptosis in a Whitefly Vector Activated by a Begomovirus Enhances Viral Transmission. *mSystems* 5 (2020).

Although the work is interesting, there is a major issue with this manuscript. Most of the confocal images used in this manuscript are not clear and are with very poor resolution. The 3D videos are not useful at all. The authors must be able to see the difference in their confocal images and in the published images which the authors kept mentioning while addressing the reviewer comments.

- We removed all 3D videos in this revision as suggested. We used 63X oil lens for all confocal experiments except for Fig1 and Fig3. Images were obtained under the maximal resolution of Zeiss LSM710. We believe we clearly showed all spots of interaction in our last revision. We noticed that the images you looked at were from the compressed files with lower pixels instead of the original images we uploaded. Please check our separately uploaded images for details. We attached Fig S5B below as an example. Some of enlarged images with mosaics because of reflecting <5 μm sample area.

Figure S5B

Line 94. Change to Whiteflies feeding on.

- Done.

Fig. 1 legend. Time course immunoblot of guts or salivary glands? Why aren't the midguts photographed at 60x? The resolution would have been better and the signals would have been more prominent if the pictures were zoomed in.

- Fig 1E represented the entire body of whitefly, and we have now made it clear (see Line 99).
- We used 20X lens in Fig 1 because we wanted to illustrate changes of the entire midgut upon virus acquisition. Also, we did use images photographed with 63X oil lens elsewhere, in which more detailed information was revealed. In this revision, we have added results from quantitative analysis by ImageJ (see below).

Line 139. Why ' was further verified using Y2H'? It's already been selected from the Y2H assays.

- The initial Y2H selected 23 putative targets from a cDNA library, some of which could be false positive. It is a routine practice to perform the second Y2H to validate selected genes individually. For this reason, we constructed a PEBP4 plasmid in yeast to determine if the CP-PEBP4 interaction is valid.

Line 140. If there is no interaction in the stringent media (QDO), isn't the interaction weak? Fig. 2A The growth is weaker even in TDO.

- Yes, it was a weak interaction in Y2H system. Multiple post-translational modifications (PTMs) have been identified in PEBP based on many previous studies. Thus, we think weak affinity could possibly exist in yeast because of the lack of PTMs. We therefore tested the interaction using other systems (*E. coli* and whitefly *in vivo*), and all results showed that the CP can interact with PEBP4. You mentioned the weak growth even in TDO. Perhaps it could be explained by low brightness of the image, because the intensity of the interaction (lane 4) should be as strong as the positive control (lane 2) in TDO.

Did the authors use X-gal to determine the interaction? If not, why?

- We did not use X-gal because we were in shortage of X-gal at that time and failed to purchase due to the social distancing policy during COVID19. To compensate for this, we did two sets of Y2H experiment (Fig2A and Table S1). Moreover, the results of other biochemical and confocal experiments in Fig 2B-2D can further confirm this interaction.

Fig. 2D The signals for both PEBP and CP are weak. The colocalization in the midgut merged panel is weak but no doubt, shows colocalization, however, in the salivary glands, what the authors are claiming to be colocalization (yellow dots), seem to be an artifact. I do not understand why the pictures are not clear. I would also suggest the authors include the colocalization using ImageJ or any other available software. If already done, please put them in the figures or as sup fig for better understanding. Also, put the intensity mapping in the sup. This will help the readers.

- Images were obtained using 63x oil lens, and the minimum resolution was no less than 800 x 800 dpi. As suggested, we quantified these colocalizations by ImageJ using Manders' Colocalization Coefficients (MCC) (Manders *et al.*, 1993). Please see Fig2D below as an example. M2 values represented the proportion of PEBP4-CP colocalization in total CP. Such information was also provided for Fig4G, 5G to support other colocalization results. We also added the quantification results in Fig1A,B, Fig3D,E, Fig4H as suggested.

Manders, E. M. M., Verbeek, F. J., & Aten, J. A. (1993). Measurement of co-localization of objects in dual-colour confocal images. *Journal of microscopy*, 169(3), 375-382.

Line 148. How do the authors claim that PEBP was abundant on the plasma membrane? Did they use any plasma membrane tracker? The claim is not rational.

- We did not use plasma membrane marker. Mammalian PEBPs have been found to directly interact with negatively charged membrane microdomains in cells (Roussel *et al.*, 1998; Vallée *et al.*, 2001). They are considered modulators of molecular interactions that control signal transduction during membrane and cytoskeleton reorganization (Schoentgen and Jonic, 2018). Similarly, structural prediction in our study revealed a transmembrane region at the N-terminus of PEBP4 (see below). Together with our Z-stack results, we conclude that PEBP4 exists in plasma membrane as other proteins in this family.

Vallée, B. S., Tauc, P., Brochon, J. C., Maget-Dana, R., Lelièvre, D., Metz-Boutigue, M. H., ... & Schoentgen, F. (2001). Behaviour of bovine phosphatidylethanolamine-binding protein with model membranes. Evidence of affinity for negatively charged membranes. *European Journal of Biochemistry*, 268(22), 5831-5841.

Roussel, G., Nussbaum, F., Schoentgen, F., Jolles, P., & Nussbaum, J. L. (1988). Immunological investigation of a 21-kilodalton cytosolic basic protein in rat brain. *Developmental Neuroscience*, 10(2), 65-74.

Schoentgen, F., & Jonic, S. (2018). PEBP1/RKIP: from multiple functions to a common role in cellular processes. *arXiv preprint arXiv:1802.02378*.

The videos of the immunolocalization aren't necessary. They don't give any better understanding. I suggest to remove them.

- Videos have been removed.

Fig. 3E. The signals would be better if taken at 60x for the midguts. The signals for Atg8-PE should also be analyzed using ImageJ for differences in intensity

- We analysed signals of both ATG8-PE and apoptosis using ImageJ as you suggested. The quantification results have been added in Fig 1, Fig 3, and Fig 4. Values in bar plots represent mean \pm SEM (*P<0.05, **P<0.01, ***P<0.001), n \geq 8.

Fig. 2B. From the western blots, it's visible that expression of PEBP is reduced in presence of TYLCV. The authors should discuss this with proper reasoning.

- We have showed this result in Fig S3 and addressed this point in the text, Line 170-172: “One scenario could be that PEBP4 protein was constantly consumed because new virions were continuously ingested by whitefly.”

Line 167. Why does the authors write that PEBP is ‘consumed’ by TYLCV? It implies TYLCV is not just binding to PEBP but is also breaking down the protein. Binding of TYLCV –CP to PEBP cannot decrease the amount of protein. There must be some transcriptional repression / gene regulation behind it. The authors must discuss this in a proper manner.

- Increased mRNA transcripts did not lead to increased PEBP4 protein abundance 48 hours after acquiring virus (Fig S3). It was likely due to elevated virus load taken in by whitefly feeding on infected hosts. We have now discussed this possibility in lines 170-172.

Apoptosis and autophagy pathways are known to regulate each other. Authors mention that PEBP acts as a molecular switch between apoptosis and autophagy, and on the other hand the whole manuscript is based on activation of both apoptosis and autophagy together. As described in Fig. 6, I suggest the authors discuss more about the crosstalk in general and how it is different here where both are activated simultaneously.

- Based on our results, we consider PEBP4 as a regulator that activates both pathways but not function as a molecular switch. We added discussion content in line 384-397: “Complex crosstalk between apoptosis and autophagy is often manifested in reciprocal antagonism. Presumably, it facilitated shaping a homeostasis during removing superfluous cells for tissue development or in determining cell fate upon virus infection. PEBPs were found to regulate apoptosis and autophagy independently, but their role in shaping apoptosis-autophagy homeostasis has been ignored. In our study, although virus load was primarily determined by autophagy, PEBP4 regulated coactivation ensured involvement of apoptosis as well. Its dynamic antagonistic effect on autophagy would prevent virus elimination when autophagy was continuously activated by virus ingestion, and thus warrant the vectoral role of whitefly. On the other side, coactivation of both apoptosis and autophagy prevented the substantial

cost on whitefly fitness, because suppressing autophagy by application of 3-MA amplified the burden of virulence.”.

REVIEWERS' COMMENTS

Reviewer #4 (Remarks to the Author):

The authors have addressed all the comments and replaced the figures with better-quality images.

The manuscript is much better and is recommended for publication. Although I have some minor changes to suggest.

I would suggest the authors remove the word 'consume' from wherever its written PEBP was consumed because the protein is not breaking down; TYLCV is binding to PEBP.

Line 137. Change to '...PEBP was re-verified using one-to-one Y2H analysis.'

And I still feel simple protein binding does not mean a reduction in the protein levels in western blots. There must be some post-transcriptional repression/ translational regulation by the whiteflies themselves. This has not been explained in lines 170-172. I would suggest the authors include this for better reasoning. Please do not write PEBP was consumed, again.

I also think writing 'PEBP was abundant ON plasma membrane' from Fig. S2 does not make any sense. The fact that PEBP has a transmembrane domain is fine, but the images in no way verify that the protein is in the membrane. No doubt, it can be said that PEBP was localized near the cell surface/membrane. Many more studies are required to prove that a protein is actually in the plasma membrane and I think that is not necessary here. Simply remove the sentence or write 'was abundant near the cell surface'.

Below is our point-by-point response to reviewers' comments. The criticisms are in RED, and our responses are in BLUE.

Reviewer #4 (Remarks to the Author):

The authors have addressed all the comments and replaced the figures with better-quality images.

The manuscript is much better and is recommended for publication. Although I have some minor changes to suggest.

I would suggest the authors remove the word 'consume' from wherever its written PEBP was consumed because the protein is not breaking down; TYLCV is binding to PEBP.

- We have replaced the word "consumed" with "hijacked" in line 131.

Line 137. Change to '...PEBP was re-verified using one-to-one Y2H analysis.'

- We have revised this sentence as proposed in line 112.

And I still feel simple protein binding does not mean a reduction in the protein levels in western blots. There must be some post-transcriptional repression/ translational regulation by the whiteflies themselves. This has not been explained in lines 170-172. I would suggest the authors include this for better reasoning. Please do not write PEBP was consumed, again.

- We have replaced the word "consumed" with "hijacked" in line 131.

I also think writing 'PEBP was abundant ON plasma membrane' from Fig. S2 does not make any sense. The fact that PEBP has a transmembrane domain is fine, but the images in no way verify that the protein is in the membrane. No doubt, it can be said that PEBP was localized near the cell surface/membrane. Many more studies are required to prove that a protein is actually in the plasma membrane and I think that is not necessary here. Simply remove the sentence or write 'was abundant near the cell surface'.

- As proposed, we have rewritten the sentence in line 122 with "near the cell surface".